# Loss of MIG-6 results in endometrial progesterone resistance via ERBB2

Jung-Yoon Yoo[1,2,9], Tae Hoon Kim[1,9], Jung-Ho Shin[3], Ryan M. Marquardt [1,4], Ulrich Müller[5], Asgerally T. Fazleabas [1], Steven L. Young [6], Bruce A. Lessey[7], Ho-Geun Yoon [8✉] & Jae-Wook Jeong [1✉]

Female subfertility is highly associated with endometriosis. Endometrial progesterone resistance is suggested as a crucial element in the development of endometrial diseases. We report that *MIG-6* is downregulated in the endometrium of infertile women with endometriosis and in a non-human primate model of endometriosis. We find ERBB2 overexpression in the endometrium of uterine-specific *Mig-6* knockout mice (*Pgr^{cre/+}Mig-6^{f/f}*; *Mig-6^{d/d}*). To investigate the effect of ERBB2 targeting on endometrial progesterone resistance, fertility, and endometriosis, we introduce *Erbb2* ablation in *Mig-6^{d/d}* mice (*Mig-6^{d/d}Erbb2^{d/d}* mice). The additional knockout of *Erbb2* rescues all phenotypes seen in *Mig-6^{d/d}* mice. Transcriptomic analysis shows that genes differentially expressed in *Mig-6^{d/d}* mice revert to their normal expression in *Mig-6^{d/d}Erbb2^{d/d}* mice. Together, our results demonstrate that ERBB2 overexpression in endometrium with MIG-6 deficiency causes endometrial progesterone resistance and a nonreceptive endometrium in endometriosis-related infertility, and ERBB2 targeting reverses these effects.

[1] Department of Obstetrics,Gynecology & Reproductive Biology, Michigan State University, College of Human Medicine, Grand Rapids, MI, USA. [2] Department of Biomedical Laboratory Science, Yonsei University Mirae Campus, Wonju, South Korea. [3] Division of Reproductive Endocrinology, Department of Obstetrics & Gynecology, Guro Hospital, Korea University Medical Center, Seoul, South Korea. [4] Cell and Molecular Biology Program, Michigan State University, East Lansing, MI, USA. [5] The Solomon H. Snyder Department of Neuroscience, Johns Hopkins University School of Medicine, Baltimore, MD, USA. [6] Department of Obstetrics and Gynecology, University of North Carolina, Chapel Hill, NC, USA. [7] Department of Obstetrics and Gynecology, Wake Forest Baptist Health, Winston-Salem, NC, USA. [8] Department of Biochemistry and Molecular Biology, Severance Medical Research Institute, Graduate School of Medical Science, Brain Korea 21 Project, Yonsei University College of Medicine, Seoul, South Korea. [9]These authors contributed equally: Jung-Yoon Yoo, Tae Hoon Kim. ✉email: yhgeun@yuhs.ac; jeongj@msu.edu

Critical for fertility, the uterine endometrium's epithelial and stromal compartments undergo dynamic hormonally controlled molecular and morphological changes to prepare for embryo implantation and development. Estrogen (E2) stimulates the proliferation of uterine epithelial cells, and progesterone (P4) suppresses E2-induced proliferation. Endometrial P4 resistance implies decreased responsiveness of target tissue to bioavailable P4[1–3]. Endometrial P4 resistance is seen in women with a nonreceptive endometrium, endometriosis, polycystic ovary syndrome (PCOS), and endometrial cancer[4–9]. Moreover, P4-induced molecular changes in the eutopic (intrauterine) endometrial tissue of women with endometriosis are either blunted or undetectable[10–12].

Endometriosis affects about 10% of all women of reproductive age, and the incidence increases to 50-60% of women with chronic pelvic pain and infertility[13,14]. While progestin-based therapies are commonly used to treat endometriosis and lead to disease regression in some women, other women with endometriosis and pelvic pain do not respond effectively to progestins[15,16]. Moreover, many P4-induced molecular changes in the eutopic endometrial tissue of women with endometriosis are either blunted or dysregulated[17,18], but an impaired P4 response is seen in the endometrium of women with endometriosis[4–6,10]. Despite knowing the effects, the molecular mechanism responsible for endometrial P4 resistance and dysregulation remains unclear. Therefore, understanding the molecular mechanisms of endometrial P4 resistance is critical.

Here, we show that the amount of mitogen inducible gene 6 (MIG-6) is decreased in endometrium from infertile women with endometriosis. We use uterine-specific Mig-6 knock-out mice to demonstrate that MIG-6 loss results in endometrial progesterone resistance via erb-b2 receptor tyrosine kinase 2 (ERBB2; also known as CD340, proto-oncogene Neu, or HER2). Our findings provide insight into the etiology of female infertility and a molecular framework useful for the design of therapeutic strategies.

## Results

**MIG-6 is decreased in endometriosis-affected endometrium**. We previously identified Mig-6 as a P4-regulated gene that mediates the ability of P4 to repress E2 action in the mouse uterus[19,20]. During the menstrual cycle, P4 amounts rise at the early secretory phase. As measured by RT-qPCR, MIG-6 expression in the human endometrium was significantly higher in the early secretory phase of the menstrual cycle than in the proliferative ($p < 0.0001$), mid secretory ($p = 0.0106$), and late secretory phase ($p = 0.0029$) (Fig. 1a), suggesting that MIG-6 is a P4-induced gene in the human endometrium as has been demonstrated in the mouse[20]. Because many P4-induced endometrial molecular changes are either blunted or eliminated in women with endometriosis[1,5,8], we examined MIG-6 expression in endometrial biopsies from infertile women with endometriosis. RT-qPCR and immunohistochemistry showed that amounts of MIG-6 mRNA ($p = 0.0079$) and protein ($p < 0.0001$) were significantly lower in the eutopic endometrium of infertile women with endometriosis compared to controls in the early secretory phase (Fig. 1, a–c). To assess how MIG-6 expression is affected by endometriosis progression, we used a baboon model[21]. Intraperitoneal inoculation with autologous menstrual effluent in female nonhuman primates results in the formation of endometriotic lesions highly similar in histomorphology to those seen in women[22]. We found that endometrial MIG-6 protein abundance was significantly reduced in baboons during the progression of endometriosis after experimental disease induction as compared to paired pre-inoculation control samples (Fig. 1d; $p < 0.0001$). These results demonstrate that reduced MIG-6 expression can be caused by the development of endometriotic lesions.

Uncovering pathophysiological mechanisms of endometriosis-related infertility with animal models requires easy identification of lesions to distinguish them from the surrounding normal tissues. With this in mind, we developed a mouse model of endometriosis using mT/mG reporters. In $Pgr^{cre/+}Rosa26^{mTmG/+}$ mice, progesterone receptor (Pgr)-positive uterine cells express mG, while Pgr-negative cells express mT (Supplementary Fig. 1a, b). Using this model, we surgically induced endometriosis in $Pgr^{cre/+}Rosa26^{mTmG/+}$ mice by inoculating autologous endometrial tissue fragments into the peritoneal cavity after 3 days of E2 treatment (Supplementary Fig. 1c). This method leads to the development of endometriotic lesions similar to those in humans without the need for ovariectomy or unopposed E2 treatment (Supplementary Fig. 1d–f). To examine the responsiveness of our endometriosis model to E2 and P4, $Pgr^{cre/+}Rosa26^{mTmG/+}$ mice induced with endometriosis were treated with vehicle, E2, or E2 + P4 for one month. While E2 treatment after endometriosis induction significantly increased the number of endometriotic lesions compared to the vehicle group, the addition of P4 suppressed the E2-induced increase in lesion number (Supplementary Fig. 1g, h; $p = 0.0077$ and $p = 0.0014$). Our mouse model thus closely mirrors human endometriosis as an E2-dependent and P4-suppressed disorder. To determine whether MIG-6 expression is dysregulated after endometriosis development in a distinct mammalian system, we examined MIG-6 levels in eutopic endometrium from $Pgr^{cre/+}Rosa26^{mTmG/+}$ mice with endometriosis. MIG-6 protein expression was significantly reduced in eutopic endometrium from the mice with endometriosis compared to the sham group (Fig. 1e; $p < 0.0001$).

**The effect of Mig-6 loss in endometriosis-related infertility**. Next, we assessed whether endometriosis in mice causes infertility by assessing implantation and decidualization success (Fig. 2a). One month after endometriosis induction, the number of implantation sites in mice with endometriosis was not changed compared to the sham group. However, 63.6% (7 out of 11) of mice with endometriosis experienced implantation failure 3 months after endometriosis development (Fig. 2b). We found that no significant correlation exists between the number of implantation sites and the number of endometriosis lesions (Supplementary Fig. 2). We next examined the impact of endometriosis on decidualization using an artificial decidualization model[23]. One month after endometriosis induction, mice with endometriosis displayed a uterine horn that responded well to artificial decidualization; however, after 3 months of endometriosis development, the mice with endometriosis exhibited a significant defect in decidual response compared to control and sham mice (Fig. 2c; $p = 0.0001$). Our result suggests that endometriosis development causes implantation failure and a defect of decidualization, as has been hypothesized in humans[5].

Having established the link between endometriosis development and MIG-6 attenuation in the eutopic endometrium, we sought to determine if MIG-6 depletion is involved in endometriotic lesion development. In a comparison of MIG-6 expression in paired ectopic and eutopic endometrial biopsies taken from women with endometriosis, MIG-6 amounts were significantly reduced in the ectopic endometrial specimens ($p = 0.0022$) (Fig. 3a, b). To assess the effect of MIG-6 deficiency in endometriosis development, we induced endometriosis in control ($Pgr^{cre/+}Rosa26^{mTmG/+}$) and $Mig-6^{d/d}Rosa26^{mTmG/+}$ mice and found that uterine MIG-6 attenuation significantly increased incidence ($p = 0.0048$) and weight of endometriotic lesions ($p = 0.0154$) (Fig. 3c, d). To address the role of MIG-6 in endometriosis-related infertility, we surgically induced endometriosis in wild-type females using endometrial fragments from

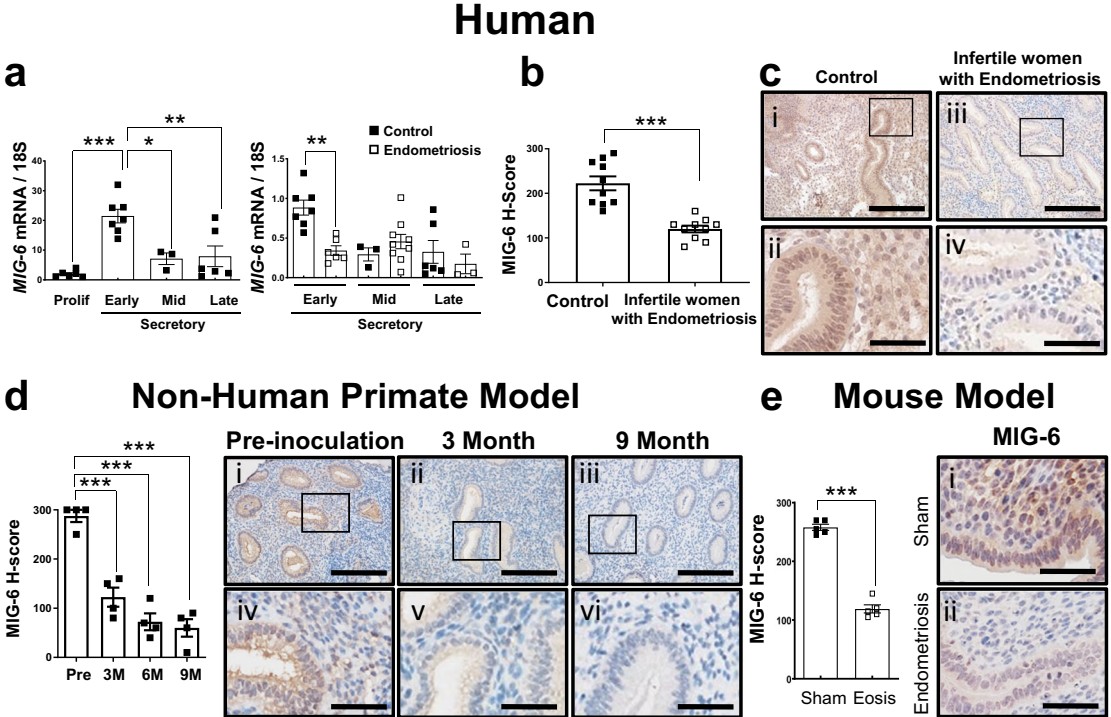

**Fig. 1 MIG-6 expression in the endometrium of women with endometriosis and nonhuman primate, baboon model. a** RT-qPCR analysis of *MIG-6* gene expression in endometrium from women with and without endometriosis during the menstrual cycle ($n = 6$ for proliferative, $n = 7$ for early secretory, $n = 3$ for mid secretory, and $n = 6$ for late secretory without endometriosis and $n = 6$ for early secretory, $n = 9$ for mid secretory, and $n = 3$ for late secretory with endometriosis). Filled square boxes represent endometrial samples from women without endometriosis, and empty square boxes represent endometrial samples from women with endometriosis. Data are represented as mean ± SEM, * $p = 0.0106$, ** $p = 0.0029$, *** $p < 0.0001$, and ** $p = 0.0079$ by Ordinary one-way ANOVA test. **b**, **c** Immunohistochemical H-score (**b**) and representative photomicrographs (**c**) of MIG-6 in the endometrium from women with endometriosis as compared to controls ($n = 10$ for each group) at early secretory phase. Data are represented as mean ± SEM, *** $p < 0.0001$ by two-tailed unpaired *t* test. Scale bars: 200 μm for (i) and (iii) and 50 μm for (ii) and (iv). **d** Immunohistochemical H-score and representative photomicrographs of MIG-6 in the endometriosis baboon model induced by intraperitoneal inoculation of menstrual endometrium during progression of endometriosis in pre-inoculation, 3, 6, and 9 months ($n = 4$ per period). Data are represented as mean ± SEM, *** $p < 0.0001$ by Ordinary one-way ANOVA test. Scale bars: 200 μm for (i), (ii), and (iii) and 50 μm for (iv), (v), and (vi). **e** Immunohistochemical H-score and representative photomicrographs of MIG-6 in the endometriosis mouse model ($n = 5$ for group). Data are represented as mean ± SEM, *** $p < 0.0001$ by two-tailed unpaired *t* test. Scale bars: 50 μm. Three independent experiments were performed for a - e with similar results. Source data are provided in the Source Data file.

donor control ($Pgr^{cre/+}Rosa26^{mTmG/+}$) and $Mig\text{-}6^{d/d}Rosa26^{mTmG/+}$ mice (Fig. 3e). One month after endometriosis induction, the number of implantation sites was significantly reduced in the mice with $Mig\text{-}6^{d/d}Rosa26^{mTmG/+}$ ectopic lesions compared to the mice with control ectopic lesions ($p = 0.002$). Furthermore, implantation sites were entirely absent from mice with $Mig\text{-}6^{d/d}Rosa26^{mTmG/+}$ ectopic lesions after 2 months of endometriosis development (Fig. 3f, g). These results demonstrate that MIG-6 attenuation in ectopic lesions increased endometriosis development and accelerated implantation failure compared to controls.

Cessation of epithelial E2-induced proliferation is essential for implantation in all eutherian mammal species studied[24,25]. In mice, abundant proliferation of epithelial and stromal cells is detectable at day 2.5 of gestation (GD 2.5). However, just before implantation, P4 inhibits epithelial proliferation and induces differentiation to an embryo receptive state[26]. Establishing uterine receptivity by sequential actions of E2 and P4 on endometrial cells is critical for successful embryo apposition, attachment, implantation, and pregnancy maintenance, and lack of sufficient E2 and P4 action can result in infertility and pregnancy loss in humans[5,7,10,27] and mice[28–31]. $Mig\text{-}6^{d/d}$ mice are infertile due to P4 resistance and implantation failure[32]. To determine whether a defect of embryo implantation is caused by an alteration in endometrial cell proliferation, we examined the expression of a

proliferation marker (Ki67) at pre-implantation (GD 3.5). Epithelial proliferation was significantly increased in the $Mig\text{-}6^{d/d}$ endometrium compared to controls ($p < 0.0001$) ($Mig\text{-}6^{f/f}$; Supplementary Fig. 3a, b). To identify the molecular explanation for the effect of MIG-6 loss on epithelial proliferation, we examined amounts of several E2 signaling molecules, including epidermal growth factor receptor (EGFR), ERBB2 and extracellular-signal-regulated kinase 1/2 (ERK1/2) at GD 3.5 in $Mig\text{-}6^{f/f}$ and $Mig\text{-}6^{d/d}$ mice. EGFR amounts were unchanged, but ERBB2 and phospho-ERK1/2 (pERK1/2) amounts were selectively increased in $Mig\text{-}6^{d/d}$ mice (Supplementary Fig. 3c, d). These results suggest MIG-6 is a negative regulator of ERBB2/ERK signaling in the pre-implantation endometrium.

**ERBB2 overexpression causes infertility in $Mig\text{-}6^{d/d}$ mice.** In order to investigate the effect of ERBB2 targeting on nonreceptive endometrium and endometriosis with $Mig\text{-}6$ deficiency, we introduced $Erbb2$ ablation in $Mig\text{-}6^{d/d}$ mice ($Mig\text{-}6^{d/d}Erbb2^{d/d}$; Supplementary Fig. 4). To address the effect of conditional $Erbb2$ knockout on the infertility phenotype of $Mig\text{-}6^{d/d}$ mice, we mated female control, $Mig\text{-}6^{d/d}$, and $Mig\text{-}6^{d/d}Erbb2^{d/d}$ ($Pgr^{cre/+}Mig\text{-}6^{f/f}Erbb2^{f/f}$) mice with wild type male mice for 6 months to determine their overall fertility. As expected, $Mig\text{-}6^{d/d}$ mice were infertile[20], but surprisingly, $Mig\text{-}6^{d/d}Erbb2^{d/d}$ exhibited normal fecundity compared to controls ($6.40 ± 0.49$ and $7.29 ± 0.29$

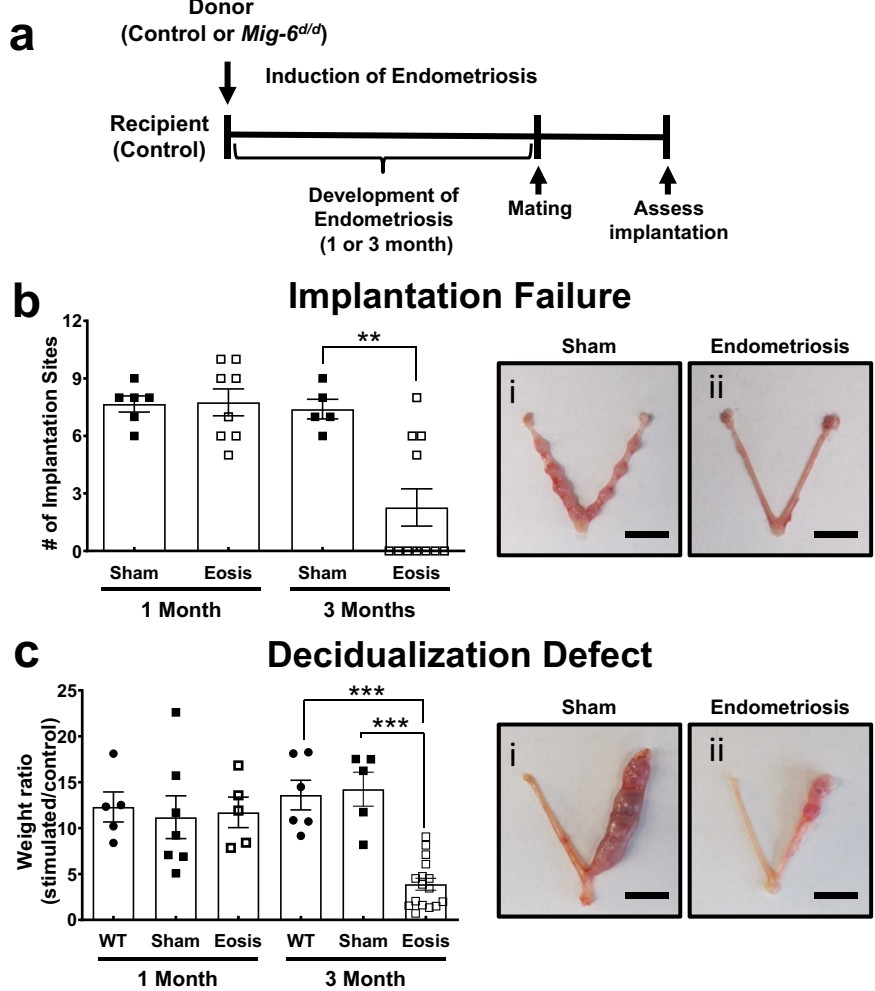

**Fig. 2 Defects of implantation and decidualization in mice with endometriosis. a** Experimental design to investigate endometriosis-related infertility. **b** Average number and uterine images of implantation sites at GD 7.5 in mice with endometriosis (Eosis) on 1 and 3 months after endometriosis induction ($n = 6$ for sham and $n = 8$ for endometriosis at 1 month and $n = 5$ for sham and for $n = 11$ endometriosis at 3 months). Data are represented as mean ± SEM, ** $p = 0.0021$ by Ordinary one-way ANOVA test. Scale bars: 1 cm. **c** Average ratio of stimulated uterine weight to control weight and uterine images of mice with endometriosis after artificially induced decidualization ($n = 5$ for wild type, $n = 7$ for sham, and $n = 5$ for endometriosis at 1 month and $n = 6$ for wild type, $n = 5$ for sham, and $n = 16$ for endometriosis at 3 months). Data are represented as mean ± SEM, *** $p = 0.0001$ by Ordinary one-way ANOVA test. Scale bars: 1 cm. Three independent experiments were performed for (**b**) and (**c**) with similar results. Source data are provided in the Source Data file.

average pups/litter, respectively; Supplementary Table 1). This is a remarkable instance of molecular targeting to correct infertility caused by endometrial P4 resistance.

To further dissect the reversal of *Mig-6*-related infertility by attenuation of *Erbb2*, we examined implantation rates. Uterine horns of *Mig-6*$^{d/d}$ mice had no grossly visible implantation sites at GD 5.5, whereas *Mig-6*$^{d/d}$*Erbb2*$^{d/d}$ mice averaged 7.00 ± 0.41 implantation sites that appeared normally spaced (Fig. 4a). Subsequent histology revealed that all embryos in *Mig-6*$^{d/d}$*Erbb2*$^{d/d}$ uteri were positioned as expected alongside the anti-mesometrial luminal epithelium, and the stromal cells had the normal decidual response surrounding the embryo (Fig. 4b). To identify the effect of additional *Erbb2* knockout on the aberrantly increased epithelial proliferation of GD 3.5 *Mig-6*$^{d/d}$ mice, we assessed Ki67 and cyclin D1 expression in *Mig-6*$^{d/d}$*Erbb2*$^{d/d}$ mice. In contrast to *Mig-6*$^{d/d}$ mice, *Mig-6*$^{d/d}$*Erbb2*$^{d/d}$ endometrial epithelial cells exhibited normal cyclin D1 and Ki67 (Fig. 4c). Since the increase of epithelial proliferation in *Mig-6*$^{d/d}$ mice is accompanied by increased E2 signaling, we investigated whether excess E2 signaling is abrogated by *Erbb2* ablation. The expression

of the E2-responsive genes mucin 1 (*Muc-1*) ($p = 0.00156$, $p = 0.0091$, and $p = 0.0029$), chloride channel calcium activated 3 (*Clca3*)) ($p < 0.0001$), lactoferrin (*Ltf*) ($p < 0.0001$), and complement component 3 (*C3*) ($p = 0.0007$, $p = 0.0003$, and $p < 0.0001$) were significantly increased in *Mig-6*$^{d/d}$ mice but restored to normal amounts in *Mig-6*$^{d/d}$*Erbb2*$^{d/d}$ mice (Fig. 4d). The same pattern was apparent for MUC1 and LTF protein amounts (Fig. 4e). However, PGR and ESR1 expression were not changed (Supplementary Fig. 5). These results imply that ERBB2 overexpression resulting from *Mig-6* attenuation causes female infertility due to a nonreceptive endometrium, and this effect may be reversed by ablation of *Erbb2*.

### *Erbb2* ablation overcomes P4 resistance in *Mig-6* mutant mice.
*Mig-6* attenuation causes endometrial P4 resistance demonstrated by P4's inability to inhibit E2-induced uterine weight gain in *Mig-6*$^{d/d}$ mice[20]. In order to determine if *Erbb2* ablation restores endometrial P4 responsiveness in *Mig-6*$^{d/d}$ mice, ovariectomized control, *Mig-6*$^{d/d}$, and *Mig-6*$^{d/d}$*Erbb2*$^{d/d}$ mice were treated with

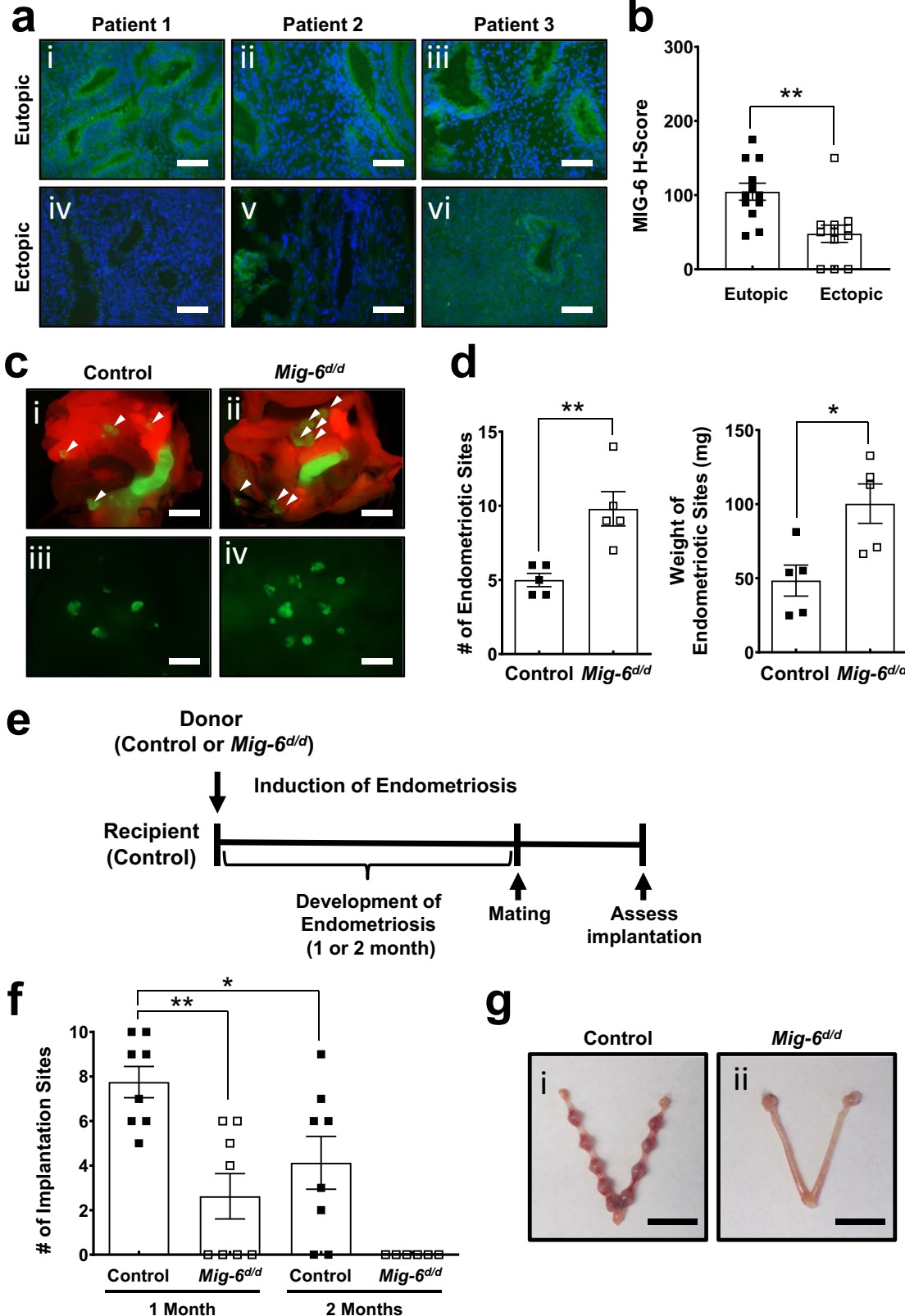

vehicle or E2 + P4 for 3 days. While *Mig-6^{d/d}* mice treated with E2 + P4 experienced significant increases in uterine weight ($p < 0.0001$), vascularization, and expression of the E2 target genes *Muc1*, *Clca3*, *Ltf*, and *C3* compared to E2 + P4 treated control mice ($p < 0.0001$), *Mig-6^{d/d}Erbb2^{d/d}* mice exhibited normal P4 responsiveness and expression of E2 target genes

(Fig. 5a–c). IHC results showed that a decrease of PGR expression in *Mig-6^{d/d}* mice was recovered in *Mig-6^{d/d}Erbb2^{d/d}* mice, but ESR1 expression was not changed (Supplementary Fig. 6). We then examined whether ERBB2 null mutation can rescue PGR-null mice's endometrial phenotypes. We have generated *Pgr^{cre/cre}Erbb2^{f/f}* and *Pgr^{cre/+}Erbb2^{f/f}* mice to determine the effect

**Fig. 3 Reduction of MIG-6 in ectopic lesions in endometriosis patients and the effect of ectopic lesions with MIG-6 deficiency on endometriosis development and embryo implantation. a, b** Decrease of MIG-6 expression in ectopic endometriotic lesions compared to eutopic endometrium from the same endometriosis patients. Representative photomicrographs (**a**) and H-score (**b**) of immunofluorescence analysis of MIG-6 in eutopic endometrium and ectopic lesions from women with endometriosis ($n = 12$ for group). Green fluorescent protein indicates MIG-6 protein expression and nuclei were counterstained with DAPI. Data are represented as mean ± SEM, ** $p = 0.0022$ by two-tailed unpaired $t$ test. Scale bars: 100 μm. **c, d** The effect of ectopic lesions with MIG-6 deficiency on endometriosis development. Endometriosis was surgically induced in control ($Pgr^{cre/+}Rosa26^{mTmG/+}$) and $Mig-6^{d/d}Rosa26^{mTmG/+}$ mice. Endometriotic lesions were visualized by GFP expression in control and $Mig-6^{d/d}Rosa26^{mTmG/+}$ mice. Fluorescence photomicrographs (**c**) and average total number (**d**) of endometriosis lesions in control and $Mig-6^{d/d}Rosa26^{mTmG/+}$ mice ($n = 5$ for group). DAPI. Data are represented as mean ± SEM, ** $p = 0.0048$ and * $p = 0.0154$ by two-tailed unpaired $t$ test. Arrowheads indicate lesions attached outside the uterus. Scale bars: 1 cm. **e–g** The effect of ectopic lesions with MIG-6 deficiency on embryo implantation. **e** Experimental design to access the effect of ectopic lesions with MIG-6 deficiency on embryo implantation. Average number (**f**) and uterine images (**g**) of implantation sites at GD 7.5 in mice with endometriosis on 1 and 3 months after endometriosis induction ($n = 8$ for control and n = 8 for $Mig-6^{d/d}$ at 1 month and $n = 8$ for control and $n = 6$ for $Mig-6^{d/d}$ at 2 months). Data are represented as mean ± SEM, * $p = 0.0366$ and ** $p = 0.0020$ by Ordinary one-way ANOVA test. Scale bars: 1 cm. Three independent experiments were performed for (**a–d**) and (**f–g**) with similar results. Source data are provided in the Source Data file.

of ERBB2 ablation in PRKO (progesterone receptor knock-out) mice. However, ERBB2 null mutation did not rescue the phenotype of implantation failure in PRKO mice (Supplementary Fig. 7). In addition, we examined the effects of E2 alone in the endometrium of normal and mutant mice. Ovariectomized control, $Mig-6^{d/d}$, $Erbb2^{d/d}$, and $Mig-6^{d/d} Erbb2^{d/d}$ mice were treated with E2 alone for 3 days. However, the uterus/body weight measurements were not different between control, $Mig-6^{d/d}$, $Erbb2^{d/d}$, and $Mig-6^{d/d} Erbb2^{d/d}$ mice after E2 treatment (Supplementary Fig. 8).

We then examined the expression of ERBB2 in the induced endometriosis of nonhuman primates. We observed increased ERBB2 in the eutopic endometrium from the same baboons over time and with endometriosis progression. There are reverse correlations between MIG-6 and ERBB2 proteins (Fig. 6a–c). We next examined the effect of $Erbb2$ ablation in the endometriosis development of $Mig-6^{d/d}$ mice and found the number and weight of endometriotic lesions were restored to control amounts by the additional ablation of $Erbb2$ (Fig. 6d, e; Supplementary Fig. 9).

Uterine $Mig-6$ ablation causes endometrial hyperplasia by 5 months of age[20]. To investigate the impact of additional $Erbb2$ knockout on endometrial hyperplasia development due to $Mig-6$ attenuation, we examined uterine weight and gross histological morphology in control, $Mig-6^{d/d}$, and $Mig-6^{d/d}Erbb2^{d/d}$ mice at 5 months of age. Uterine weight was significantly decreased in $Mig-6^{d/d}Erbb2^{d/d}$ mice when compared to $Mig-6^{d/d}$ mice ($p < 0.0001$), and histological analysis revealed that $Mig-6^{d/d}Erbb2^{d/d}$ mice did not develop endometrial hyperplasia (Supplementary Fig. 10). These results demonstrate that all known female reproductive phenotypes caused by knocking out uterine $Mig-6$ are restored to baseline by also knocking out $Erbb2$.

To identify the signaling pathways that $Mig-6$ regulates at pre-implantation, we performed transcriptomic analysis on the uteri from control, $Mig-6^{d/d}$, and $Mig-6^{d/d}Erbb2^{d/d}$ mice at GD 3.5. We found 1,022 and 771 increased or decreased transcripts, respectively, in the $Mig-6^{d/d}$ uterus as compared with controls (Fig. 7a and Supplementary Table 2). Remarkably, 1,722 of the altered genes (96.04 %) in $Mig-6^{d/d}$ mice reverted to their normal expression levels in $Mig-6^{d/d}Erbb2^{d/d}$ mice. Pathway analysis showed that major altered pathways in the $Mig-6^{d/d}$ uterus included cell-cycle control and DNA replication. P4 blocks E2-induced DNA synthesis by inhibiting replication licensing including mini-chromosome maintenance (MCM) proteins[33,34] which have a role in both the initiation and elongation phases of eukaryotic DNA replication as part of the MCM complex[35,36]. Fifteen genes associated with cell cycle and DNA replication were significantly changed in the $Mig-6^{d/d}$ uterus (Supplementary Table 3). RT-qPCR analysis confirmed that the additional knockout of $Erbb2$ in $Mig-6^{d/d}$ mice restored dysregulated cell-

cycle control and DNA-replication-related gene transcripts to normal (Fig. 7b). IHC results showed that at the protein level as well, aberrant overexpression of MCM2 and MCM6 occurred in $Mig-6^{d/d}$ mice at the pre-implantation stage but reverted to normal in $Mig-6^{d/d}Erbb2^{d/d}$ mice (Fig. 7c). A similar action can be ascribed to P4 and E2 in the human endometrial epithelium, since a loss of MCM proteins occurs in the secretory phase, and P4 dominates this phase of the menstrual cycle[37]. Additionally, aberrant overexpression of MCM2 and MCM6 may cause abnormal epithelial proliferation and nonreceptive endometrium in infertile women with endometriosis[38]. Two Kruppel-like transcription factors (KLFs) are implicated in E2 and P4 modulation of uterine proliferation[38]. $Klf4$ expression is increased by E2 and promotes DNA replication, whereas $Klf15$ is increased by P4 and inhibits growth via regulation of $Mcm2$[38]. The expression of KLF4 was significantly increased in $Mig-6^{d/d}$ mice compared to control mice while the expression of KLF15 was decreased in $Mig-6^{d/d}$ mice ($p = 0.0008$), and the amounts reverted to normal in $Mig-6^{d/d}Erbb2^{d/d}$ mice ($p = 0.0002$) (Fig. 7b, d). These results suggest that $Erbb2$ overexpression due to $Mig-6$ ablation causes E2-induced epithelial proliferation and P4 resistance by disrupting cell cycle regulation.

## Discussion

This study reveals the attenuation of MIG-6 in eutopic endometrium from infertile women with endometriosis compared to controls. MIG-6 expression was higher in human endometrium from the early secretory phase than in endometrium from the proliferative phase. Because of the complexity and dynamic nature of implantation, the molecular processes underlying these changes are poorly understood. Improving fertility rates requires unraveling molecular mechanisms of implantation. However, how regulation occurs between P4 and E2 is still not fully understood[39,40], which is a critical barrier to better therapies for infertility. Amounts of MIG-6 mRNA and protein were lower in the eutopic endometrium of infertile women with endometriosis compared to controls in the early secretory phase. These results suggest that MIG-6 is a P4-responsive gene in human endometrium as in the mouse[20], and MIG-6 loss may result in a nonreceptive endometrium in endometriosis-related infertility.

Nonhuman primates are advantageous for studying endometriosis because they are phylogenetically similar to humans[41–43]. Intraperitoneal inoculation with autologous menstrual effluent results in the formation of endometriotic lesions similar in histology and morphology to those seen in women[22]. Paired sequential analysis showed MIG-6 protein amounts were decreased in the eutopic endometrium of baboons during the progression of endometriosis as compared to pre-inoculation control. Furthermore, MIG-6 protein expression was reduced in

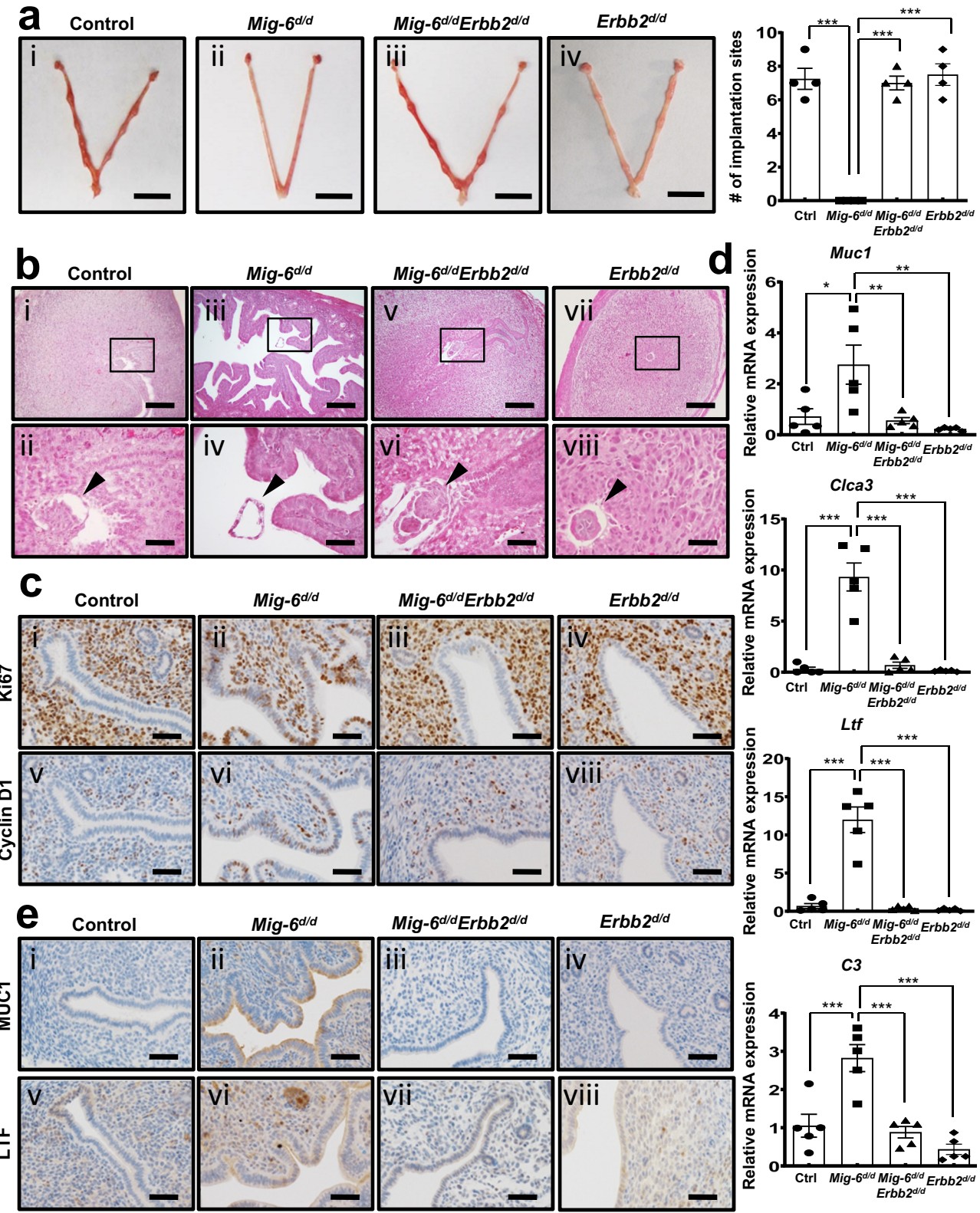

eutopic endometrium from the mice with endometriosis compared to the sham group. This result demonstrated that reduced MIG-6 expression is associated with endometriosis development.

We developed a mouse model of endometriosis based on $Pgr^{cre/+}$ and $mT/mG$ reporters that produces endometriotic lesions highly similar to those in humans. A mouse model in which excised human endometrial fragments are introduced into

the peritoneum of immunocompromised mice is widely used, but it is limited by the lack of a normal immune system, which is thought to be important in endometriosis pathophysiology[44-46]. In contrast, the mouse model of induced endometriosis is a versatile model that has been used to study how the immune system[47], hormones[48,49], and environmental factors[50,51] affect endometriosis. The availability of a large number of transgenic

**Fig. 4 Rescue of implantation defect and recovery of aberrant activated epithelial cells proliferation and ESR signaling in *Mig-6^{d/d}* mice by *Erbb2* double ablation. a** Uteri of control, *Mig-6^{d/d}*, *Mig-6^{d/d}Erbb2^{d/d}*, and *Erbb2^{d/d}* mice and number of implantation sites at GD 5.5 ($n = 4$ for each genotype). Data are represented as mean ± SEM, * $p < 0.0001$ by Ordinary one-way ANOVA test. Scale bars: 1 cm. **b** Hematoxylin and eosin (H&E) staining in paired endometrium of control, *Mig-6^{d/d}*, *Mig-6^{d/d}Erbb2^{d/d}*, and *Erbb2^{d/d}* mice at GD 5.5. Arrowheads indicate embryos. Scale bars: 200 μm for (i), (iii), (v), and (vii) and 50 μm for (ii), (iv), (vi), and (viii). **c** Immunohistochemistry analysis of Ki67 and Cyclin D1 in the endometrium of control, *Mig-6^{d/d}*, *Mig-6^{d/d}Erbb2^{d/d}*, and *Erbb2^{d/d}* mice at GD 3.5. Scale bars: 50 μm. **d, e** RT-qPCR analysis of *Muc1* (* $p = 0.00156$, ** $p = 0.0091$, and ** $p = 0.0029$), *Clca3*(*** $p < 0.0001$), *Ltf* (*** $p < 0.0001$), and *C3* (*** $p = 0.0007$, *** $p = 0.0003$, and *** $p < 0.0001$). Data are represented as mean ± SEM and analyzed by Ordinary one-way ANOVA test. (**d**) and immunohistochemistry analysis of MUC1 and LTF (**e**) as epithelial ESR1 target genes in the uterus of control, *Mig-6^{d/d}*, *Mig-6^{d/d}Erbb2^{d/d}*, and *Erbb2^{d/d}* mice at GD 3.5 ($n = 5$ for each genotype). Scale bars: 50 μm. Three independent experiments were performed for a - e with similar results. Source data are provided in the Source Data file.

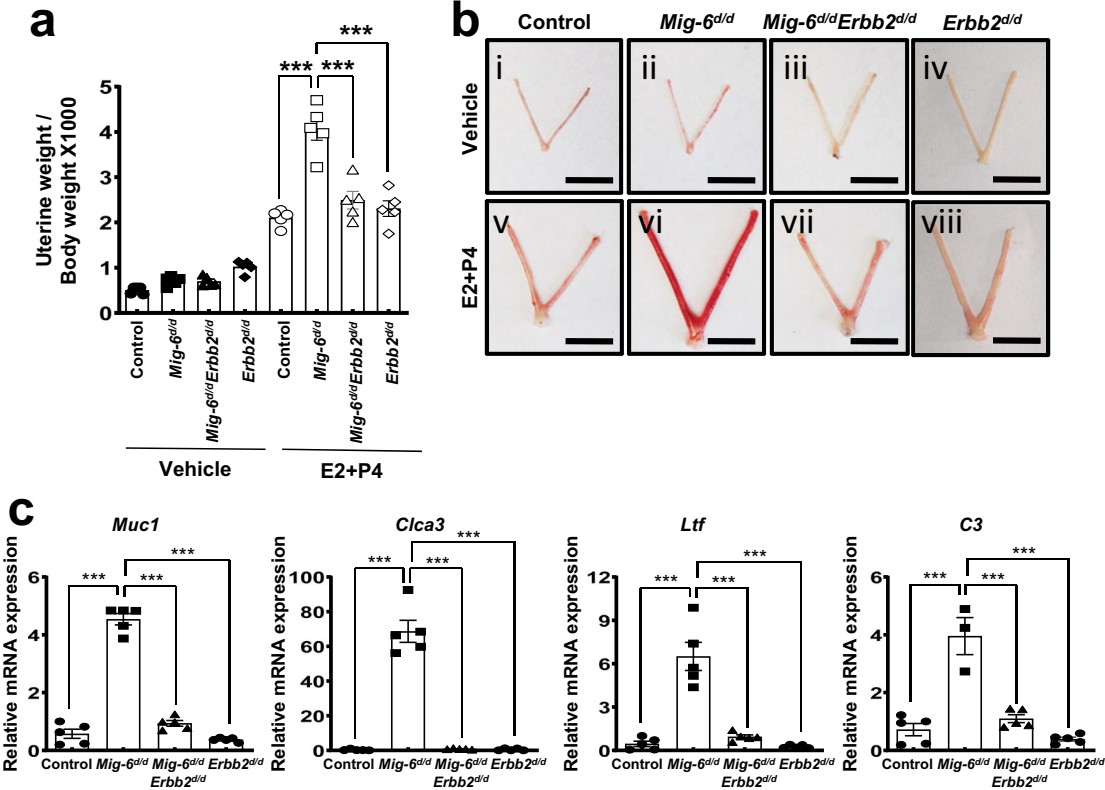

**Fig. 5 Rescue of steroid hormone dysregulation in *Mig-6^{d/d}* mice by *Erbb2* double ablation. a, b** Ratio of uterine weight to body weight (**a**) and uteri (**b**) of control, *Mig-6^{d/d}*, *Mig-6^{d/d}Erbb2^{d/d}*, and *Erbb2^{d/d}* mice treated with vehicle or E2 + P4 for 3 days ($n = 5$ per genotype). Data are represented as mean ± SEM, *** $p < 0.0001$ by Ordinary one-way ANOVA test. Scale bars: 1 cm (**c**), RT-qPCR analysis of epithelial ESR1 target genes expression, *Muc1*, *Clca3*, *Ltf*, and *C3* in the uteri of control, *Mig-6^{d/d}*, *Mig-6^{d/d}Erbb2^{d/d}*, and *Erbb2^{d/d}* mice treated with E2 + P4 for 3 days ($n = 5$ per genotype). Data are represented as mean ± SEM, *** $p < 0.0001$ by Ordinary one-way ANOVA test. Three independent experiments were performed for (**a–c**) with similar results. Source data are provided in the Source Data file.

mice in which specific genes can be either eliminated or over-expressed make this induced endometriosis model ideal for studying specific pathways in the development and progression of endometriosis and other diseases[46]. However, current mouse models of endometriosis that involve ovariectomy and E2 treatment are impractical for studies of physiological functions that require natural fluctuations in ovarian steroid hormones, such as fertility. On the other hand, our mouse model alleviates the need to apply ovariectomy and E2 treatment to enlarge endometriotic lesions because fluorescence reporter genes allow us to visualize in vivo and in real-time endometriotic lesions like those found in humans. Moreover, similarities between our mouse model and human endometriosis include: (1) development and progression of disease; (2) steroid hormone regulation; (3) fertility defect with implantation failure; and (4) P4 resistance in endometrium with *Mig-6* deficiency. Furthermore, the fluorescence reporters enable

us to quantitatively examine endometriotic lesions in these mice more accurately and easily than in prior models.

Because *Mig-6^{d/d}* mice have a fertility defect[20,32], we applied a syngeneic mouse model to examine the effect of endometriotic lesions with *Mig-6* ablation on the eutopic endometrium. Several groups have used syngeneic mouse models of endometriosis, in which the uterus of one mouse is removed, minced and injected intraperitoneally into recipient mice[46]. Syngeneic murine models have several potential advantages over the rodent surgical model: (1) peritoneal seeding of uterine fragments is more similar to retrograde menstruation in women; (2) either the donor or recipient animal can receive therapeutic intervention or be otherwise manipulated prior to induction of disease; and (3) a large number of transgenic mice in which specific genes can be either eliminated or overexpressed are available. These advantages make syngeneic murine models ideal for studying the role of

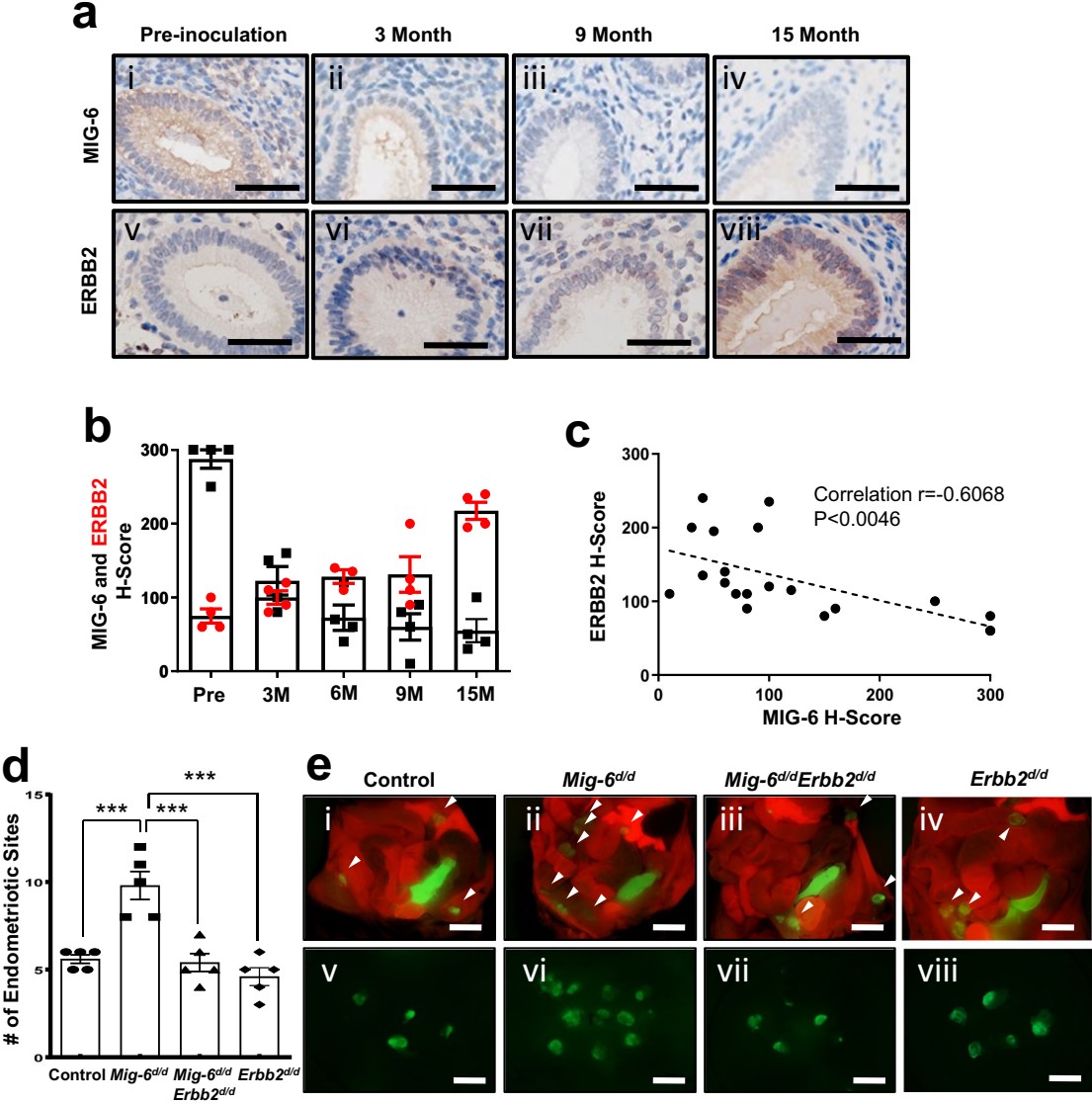

**Fig. 6 The reverse correlations between MIG-6 and ERBB2 proteins in the endometriosis baboon model. a–c** Immunohistochemical representative photomicrographs (**a**) and H-score (**b**) of MIG-6 (black) and ERBB2 (red) in the endometriosis baboon model induced by intraperitoneal inoculation of menstrual endometrium during progression of endometriosis in pre-inoculation, 3, 6, 9, and 15 months (*n* = 4 per period). Scale bars: 50 μm. **c** The reverse correlations between MIG-6 and ERBB2 proteins (Spearman correlation coefficient *r* = −0.6068, The two-tailed *p* value is 0.0046). **d, e** Average total number (**d**) and representative fluorescence photomicrographs (**e**) of endometriotic sites in control, *Mig-6$^{d/d}$*, *Mig-6$^{d/d}$Erbb2$^{d/d}$*, and *Erbb2$^{d/d}$* mice by induced endometriosis based on *mT/mG* mice (*n* = 5 per genotype). Endometriotic lesions were visualized by GFP expression in the outside of uterus. White arrow indicates endometriotic lesions. Data are represented as mean ± SEM, *** *p* = 0.0003, *** *p* = 0.0002, and *** *p* < 0.0001 by Ordinary one-way ANOVA test. Scale bars: 1 cm. Three independent experiments were performed for a - e with similar results. Source data are provided in the Source Data file.

specific pathways in the development and progression of endometriosis and other diseases.

P4 is absolutely required for uterine implantation, decidualization, and maintenance of pregnancy[8,52]. How endometriosis contributes to infertility remains elusive, although P4 resistance is likely involved[1]. P4 resistance is seen in the endometrium of infertile women with endometriosis, and *Mig-6$^{d/d}$* mice exhibit P4 resistance by the inability of P4 to inhibit E2-induced uterine weight gain[20]. We demonstrate that MIG-6 mediates P4 inhibition of E2-induced cell proliferation by inhibition of ErbB2-ERK signaling. MIG-6 plays an important role in inhibiting epithelial cell proliferation and facilitating implantation. Epithelial cell proliferation and cyclin D1 amounts were higher in the epithelial cells of *Mig-6$^{d/d}$* mice, whereas both *Mig-*

*6$^{d/d}$Erbb2$^{d/d}$* and control mice lacked elevated cyclin D1 amounts and epithelial cell proliferation. These results suggest that MIG-6 is a negative regulator of ErbB2 and suppresses E2-induced epithelial cell proliferation at the pre-implantation stage.

Previously we identified MIG-6 as a target of PGR in the uterus necessary to retain P4 responsiveness for endometrial homeostasis and successful embryo implantation[20,32]. Past studies also showed that MIG-6 binds to ERBB2 and inhibits its signaling in vitro[53–55]. In the present study, we delineate the action of PGR through MIG-6 to suppress ERBB2 expression in vivo in the uterus. Interestingly, we observed a reduction of PGR expression in *Mig-6$^{d/d}$* endometrial epithelium and stroma compared to controls after E2 + P4 treatment for 3 days, and this change was rescued by the additional ablation of *Erbb2*. This finding raises

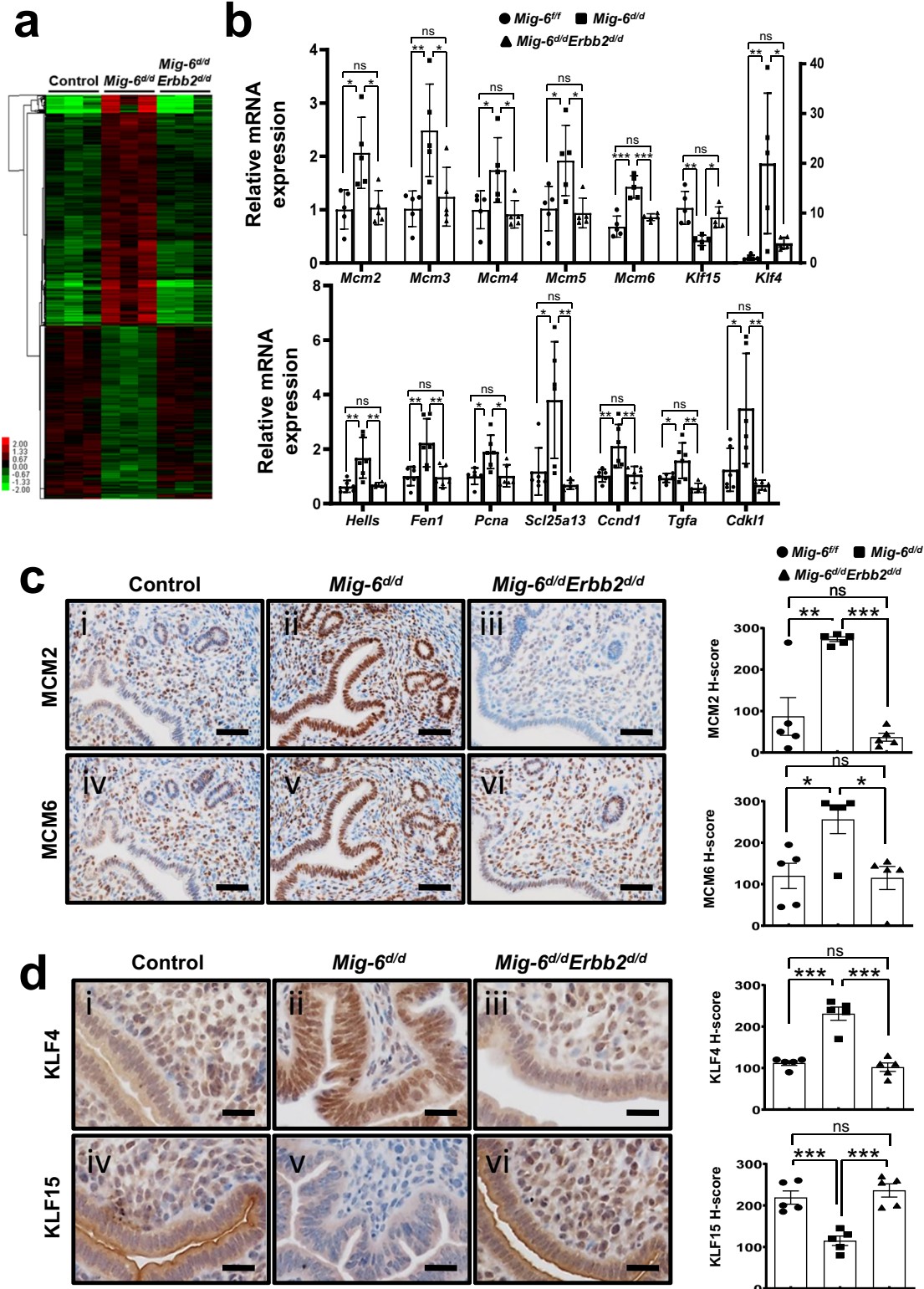

the possibility of a positive feedback loop where repression of *Erbb2* by PGR-induced MIG-6 expression is necessary to maintain PGR expression in the presence of combined E2 + P4 treatment. Therefore, the negative consequences of MIG-6 loss in the endometrium of infertile women with endometriosis may include further loss of PGR and P4-responsiveness. However, we did not observe the loss of PGR in *Mig-6^{d/d}* mice at GD3.5, implying that this feedback mechanism may not be active during

natural pregnancy conditions. In addition, it is interesting in light of known epithelial-stromal crosstalk mechanisms regulating endometrial PGR[56] that ERBB2 overexpression in *Mig-6^{d/d}* uteri occurs primarily, though not only, in the epithelium. However, PGR reduction after E2 + P4 treatment occurs in both the epithelial and stromal compartments of *Mig-6^{d/d}* uteri, and it is rescued in both compartments by *Erbb2* deletion. These data imply the presence of epithelial to stromal communication in the

**Fig. 7 Recovery of gene expression (Microarray) in *Mig-6$^{d/d}$* mice by *Erbb2* double ablation. a** Clustering analysis of *Mig-6* dependent regulated genes in uteri of control, *Mig-6$^{d/d}$*, and *Mig-6$^{d/d}$Erbb2$^{d/d}$* mice at GD 3.5. The extent of gene expression changes is represented by a green-red color scale (green: low expression and red: high expression). **b** RT-qPCR analysis of transcript amounts of *Mig-6* dependent regulated genes, *Mcm2* (* $p = 0.0108$ and * $p = 0.0133$), *Mcm3* (* $p = 0.0210$ and ** $p = 0.0075$), *Mcm4* (* $p = 0.0459$ and * $p = 0.0266$), *Mcm5* (* $p = 0.0286$ and * $p = 0.0176$), *Mcm6* (*** $p = 0.0005$ and *** $p < 0.0001$), *Klf15* (* $p = 0.0219$ and ** $p = 0.0022$), *Klf4* (* $p = 0.0232$ and ** $p = 0.0085$), *Hells* (** $p = 0.0029$ and ** $p = 0.0047$), *Fen1* (** $p = 0.0074$ and ** $p = 0.0059$), *Pcna* (* $p = 0.0108$ and * $p = 0.0121$), *Scl25a13* (* $p = 0.0104$ and ** $p = 0.0029$), *Ccnd1* (** $p = 0.0056$ and ** $p = 0.0072$), *Tgfa* (* $p = 0.0364$ and ** $p = 0.0016$), and *Cdkl1* (* $p = 0.0189$ and ** $p = 0.0041$) in uteri of control, *Mig-6$^{d/d}$*, and *Mig-6$^{d/d}$Erbb2$^{d/d}$* mice at GD 3.5 ($n = 5$ per each genotype of upper graph and $n = 6$ per each genotype in bottom graph). Data are represented as mean ± SEM and analyzed by Ordinary one-way ANOVA test. **c** Immunohistochemistry analysis of MCM2 (** $p = 0.0011$ and *** $p = 0.0001$) and MCM6 (* $p = 0.0227$ and * $p = 0.00184$) in the uterus of control, *Mig-6$^{d/d}$*, and *Mig-6$^{d/d}$Erbb2$^{d/d}$* mice at GD 3.5 ($n = 5$ per each genotype). Data are represented as mean ± SEM and analyzed by Ordinary one-way ANOVA test. Scale bars: 50 μm. **d** Immunohistochemistry analysis of KLF4 (*** $p < 0.0001$) and KLF15 (*** $p = 0.0008$ and *** $p = 0.0002$) in the uterus of control, *Mig-6$^{d/d}$*, and *Mig-6$^{d/d}$Erbb2$^{d/d}$* mice at GD 3.5 ($n = 5$ per each genotype). Data are represented as mean ± SEM and analyzed by Ordinary one-way ANOVA test. Scale bars: 25 μm. Three independent experiments were performed for (**b**-**d**) with similar results. Source data are provided in the Source Data file.

---

feedback from ERBB2 overexpression to PGR abrogation. Since PGR regulation in human and nonhuman primate endometrial tissue is not identical to mouse tissue[57,58], we cannot be certain whether this positive feedback mechanism is conserved between species. However, our data from a baboon endometriosis model shows a significant inverse correlation between MIG-6 and ERBB2 in the eutopic endometrium. As MIG-6 decreases with endometriosis progression, ERBB2 increases, which supports the relevance of this pathway to uterine dysfunction in primates.

We evaluated the potential therapeutic value of ErbB2 as a target for correcting endometrial P4 resistance in infertility. In our transcriptomic analysis in *Mig-6$^{d/d}$* and *Mig-6$^{d/d}$Erbb2$^{d/d}$* mice at GD 3.5, altered genes in *Mig-6$^{d/d}$* mice reverted to their normal expression amounts in *Mig-6$^{d/d}$Erbb2$^{d/d}$* mice. Pathway analysis using Ingenuity Systems Software showed that major altered pathways in the *Mig-6$^{d/d}$* uterus included cell-cycle control and DNA replication. Dr. Pollard's group showed that P4 blocks E2-induced DNA synthesis by inhibiting replication licensing including mini-chromosome maintenance (MCM) proteins[33,34]. The MCM complex has a role in both the initiation and elongation phases of eukaryotic DNA replication[35,36,59]. The overlap of genes associated with the cell cycle and DNA replication between the Pollard group's microarray results and ours is striking. In the uterine epithelium, E2 stimulates the expression of MCMs, while P4 decreases transcripts of MCM2 through MCM6[33,60]. Our IHC results showed aberrant overexpression of MCM2 and MCM6 in *Mig-6$^{d/d}$* mice at the pre-implantation stage. A similar action can be ascribed to P4 and E2 in the human endometrial epithelium, since a loss of MCM proteins occurs in the secretory phase, and P4 dominates this phase of the menstrual cycle[7,37]. However, aberrant overexpression of MCM2 and MCM6 may cause abnormal epithelial proliferation and non-receptive endometrium in infertile women with endometriosis[38]. Our results regarding MIG-6 and ERBB2/ESR1 signaling in regulating uterine function in response to hormonal signals will bring insight into uterine pathophysiology and likely lead to therapies for endometrial diseases. Figure 8 depicts our proposed molecular mechanisms of MIG-6 function in endometrial P4 resistance and associated infertility and how P4 responsiveness and fertility can be restored. However, further pathways and mechanisms precisely responsible for subfertility in women with endometriosis still need to be established. Deeper inquiry into endometrial epithelial-stromal crosstalk between ErbB2/ERK/ESR1 and PGR/MIG-6 signaling pathways will be of major importance to understanding infertility and endometriosis.

In summary, our findings reveal that attenuation of MIG-6 occurs both in endometriotic lesions and in the endometriosis-effected eutopic endometrium. Evidence from mice indicates that loss of *Mig-6* in endometriotic lesions promotes their

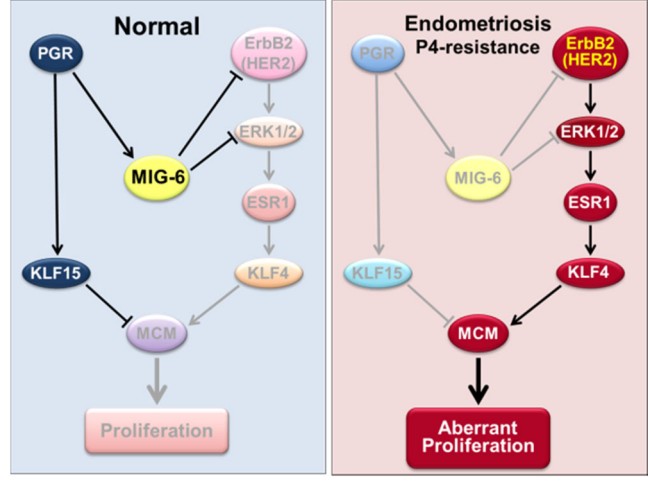

**Fig. 8 Molecular mechanisms of MIG-6 function in the uterus.** MIG-6 mediates P4 inhibition of E2 signaling by inhibiting ErbB2-ERK signaling (Left side), and the attenuation of MIG-6 leads to ErbB2-ERK activation, proliferation of uterine epithelial cells, and eventually to endometriosis and infertility (Right side).

development and accelerates endometriosis-related infertility, while loss of *Mig-6* in the eutopic endometrium causes infertility due to defects in implantation and endometrial receptivity. We found that increased epithelial proliferation caused by *Mig-6* loss is caused by E2 through the ERBB2/ERK pathway (Fig. 8). However, targeting *Erbb2* can reverse all apparent female reproductive defects caused by *Mig-6* loss including endometrial hyperplasia, infertility, and endometriosis lesion development. Attenuation of *Mig-6* causes the inability of P4 to properly control the cell cycle and inhibit E2-induced aberrant epithelial proliferation that results from increases in MCMs. However, counteracting the overexpression of *Erbb2* restores normal gene expression patterns, providing a molecular explanation for the rescue of normal reproductive function. These findings not only elucidate a critical pathway for understanding the hormonal control of normal uterine physiology, but they also provide the potential for treatment strategies for the uterine disease.

## Methods

**Study design.** The main objective of this study was to evaluate the role of MIG-6 in endometrial P4 resistance. First, the expression of MIG-6 was assessed in eutopic endometrium of infertile women with endometriosis compared to fertile women. To determine whether endometriosis affects MIG-6 expression, we examined MIG-6 expression in a nonhuman primate and mouse model of endometriosis. Subsequently, we identified ERBB2 as a MIG-6 target and evaluate the impact of *Erbb2* ablation on the infertility and endometrial P4 resistance of *Mig-6$^{d/d}$* mice. Finally,

transcriptomic analysis was applied to dissect the molecular mechanisms of *Mig-6* in the uterus. The control and treatment groups and the number of biological replicates (sample sizes) for each experiment are specified in the figure legends. Animal numbers for each study type were determined by the investigators on the basis of previous experience with the standard disease models that were used or from pilot studies. Animals were randomly allocated to the control and treatment groups and housed together to minimize environmental differences and experimental bias. Analysis of endpoint readouts was carried out in a blinded fashion.

**Ethics statement**. The institutional review boards of Michigan State University, Greenville Health System, and University of North Carolina approved this study. The Institutional Animal Care and Use Committee at Michigan State University approved all experiments relating to mice. The Institutional Animal Care and Use Committees of both the University of Illinois at Chicago and Michigan State University approved the endometriosis baboon animal model.

**Human endometrium samples**. The human endometrial samples used to examine MIG-6 expression patterns were obtained from Michigan State University's Center for Women's Health Research Female Reproductive Tract Biorepository, the University of North Carolina, and the Greenville Hospital System in accordance with the guidelines set by the Institutional Review Boards of Michigan State University (Grand Rapids, MI), the University of North Carolina (Chapel Hill, NC), and Greenville Health System (Greenville, SC), respectively. Written informed consent was obtained from all participants. The study design and conduct complied with all relevant regulations regarding the use of human study participants and was conducted in accordance with the criteria set by the Declaration of Helsinki. For experiments examining *MIG-6* mRNA expression throughout the menstrual cycle, endometrial samples were analyzed from 22 cycling premenopausal women (reproductive age; 18–40) without endometriosis (*n* = 6 proliferative, *n* = 7 early secretory, *n* = 3 mid secretory, and *n* = 6 late secretory) and from 20 cycling premenopausal women (reproductive age; 18–40) with endometriosis (*n* = 6 early secretory, *n* = 9 mid secretory, and *n* = 3 late secretory). Control endometrial tissues were laparoscopically negative for endometriosis and had not been on any hormonal therapies for at least three months prior to surgery. Histologic dating of endometrial samples was done on the basis of the criteria of Noyes et al.[61] and confirmed by subsequent histopathological examination by an experienced fertility specialist (B.A.L.). Normally cycling women without infertility who were free of hormones for at least 60 days served as fertile controls. Tubal ligation was a source for normal subjects as it allows us to rule out endometriosis laparoscopically. All endometriosis-related infertility cases were undergoing endometrial sampling prior to surgical removal of endometriosis. All patients did not have uterine leiomyoma and adenomyosis. To investigate MIG-6 amounts in the endometrium from women, 10 control and 10 eutopic endometrium with endometriosis were used. To compare MIG-6 amounts in the eutopic endometrium and ectopic lesions of women with endometriosis, each of 12 samples were used. All women with endometriosis were infertile. Samples used for immunohistochemistry were fixed in 10% buffered formalin prior to embedding in paraffin wax.

**Animals and tissue collection**. Mice were maintained in a designated animal care facility according to Michigan State University's Institutional Guidelines for the care and use of laboratory animals. All mouse procedures were approved by the Institutional Animal Care and Use Committee of Michigan State University. All housing and breeding were done in a designated animal care facility at Michigan State University with controlled humidity and temperature conditions and a 12 h light/dark cycle. Access to water and food (Envigo 8640 rodent diet) was ad libitum. Mice utilized for experiments were 8 to 12 weeks old mice from mixed background C57BL/6 and 129P2/OlaHsd strains. No statistical method was used to pre-select the sample size. Animal numbers for each study type were determined by the investigators on the basis of our previous results[62,63] with the standard disease models that were used or from pilot studies. For all animal studies, animals were randomly distributed among different conditions by the investigator as the animals did not show any size or appearance differences at the onset of the experiments. No animals were excluded, and the investigator was not blinded to group allocation during the experiment. *Erbb2* conditional knockout mice were generated by crossing *Pgr^cre/+Mig-6^f/f* with *Erbb2^f/f* mice (*Pgr^cre/+Mig-6^f/fErbb2^f/f*; *Mig-6^d/dErbb2^d/d*). Pregnant uterine samples were obtained by mating control (*Mig-6^f/f* or *Mig-6^f/fErbb2^f/f*), *Mig-6^d/d*, and *Mig-6^d/dErbb2^d/d* female mice with C57BL/6 male mice the morning of a vaginal plug designated as day 0.5 of gestation (GD 0.5). Mice were sacrificed at GD 3.5 and 5.5. For the study of steroid hormone regulation, control, *Mig-6^d/d*, *Erbb2^d/d*, and *Mig-6^d/dErbb2^d/d* mice at 6 weeks of age were ovariectomized. Two weeks post-surgery, ovariectomized mice were injected with vehicle (sesame oil; Veh) or estradiol (0.1 µg/mouse; E2) plus progesterone (1 mg/mouse; P4) for 3 days and euthanized at 6 h after injection. For the fertility studies, adult female control, *Mig-6^d/d* and *Mig-6^d/dErbb2^d/d* mice were placed with wild-type C57BL/6 male mice. The mating cages were maintained for 6 months and the number of litters and pups born during that period was recorded. Uterine tissues were then immediately processed at the time of dissection and either fixed with 4% (vol/vol) paraformaldehyde for histology or immunohistochemistry or snap frozen and stored at −80 °C for RNA/protein extraction.

**Induction of endometriosis**. For baboon uterine samples, endometriosis was induced by intraperitoneal inoculation of menstrual endometrium on two consecutive menstrual cycles and harvested using laparotomy via endometriectomy from four female baboons as previously described[64]. For mouse uterine samples, 8-weeks-old female mice which have conditional double-fluorescent Cre reporter gene (*Pgr^cre/+Rosa26^mTmG*, *Pgr^cre/+ Mig-6^f/fRosa26^mTmG*, *Pgr^cre/+ Erbb2^f/fRosa26^mTmG* and *Pgr^cre/+ Mig-6^f/fErbb2^f/f Rosa26^mTmG*) were injected with 1 µg/ml of E2 per a day at three times and had a surgical procedure to induce endometriosis. Under anesthesia, a midline abdominal incision was made to expose the uterus in female mice, and one of uterine horn was removed. In a Petri dish containing phosphate-buffered saline (PBS; pH 7.5), the uterine horn was opened longitudinally with scissors. The excised uterine horn was cut into small fragments of about 1 mm³, and then injected back into the peritoneum of the same mouse. The abdominal incision and wound were closed with sutures and skin was closed with surgical wound clips, respectively. After a designated time, the mice were euthanized, and endometriosis-like lesions were removed using a fluorescence microscope and counted.

**Endometriosis-related infertility analysis**. Endometriosis was induced in 8-week-old control female mouse recipients (fertile) receiving endometrial fragments from donor control (*Pgr^cre/+Rosa26^mTmG*) or *Mig-6^d/d Rosa26^mTmG* mice. A sham surgery group was included as a control. After 1, 2, and 3 months post endometriosis induction, the mice with endometriotic lesions of control or *Mig-6^d/d* were mated with wild-type male mice and then collected at GD 7.5.

**RNA isolation and microarray analysis**. Total RNA was extracted from the uterine tissues using the RNeasy Total RNA Isolation Kit (Qiagen, Valencia, CA). NanoDrop was used to determine RNA purity and for an initial estimate of RNA concentration. All RNA samples were analyzed with a Bioanalyzer 2100 (Agilent Technologies, Wilmington, DE) to confirm sample concentration and purity (RIN > 8.0; concentration 100-200 ng/ul) before microarray hybridization. RNA was pooled from the uteri of more than three mice per genotype at GD 3.5 and microarray analysis was performed using GeneChip® Mouse Genome 430 2.0 Arrays (Affymetrix) as described previously[65] (Gene Expression Omnibus accession code GSE138185). Array data were analyzed using Bioconductor for quantile normalization. We selected aberrantly expressed genes in the uteri of control, *Mig-6^d/d* and *Mig-6^d/dErbb2^d/d* mice at GD 3.5 using a two-sample comparison according to significant fold change greater than 1.5. Aberrantly expressed genes were classified with canonical pathway analyzed by QIAGEN Ingenuity Pathway Analysis (IPA).

**Reverse transcription—quantitative PCR**. The complementary DNAs (cDNAs) were synthesized with MMLV Reverse Transcriptase (Invitrogen Crop) according to the manufacturer's instructions. RT-qPCR was performed on cDNA to assess the expression of genes of interest with SYBR Green (Bio-Rad) or TaqMan primers (Applied Biosystems). Experimental gene expression data were normalized against the housekeeping gene, 18S ribosomal RNA. Analysis of mRNA expression was first undertaken by the standard curve method, and results were corroborated by cycle threshold values assessing gene expression. Primer sequences used in these studies are shown in Supplementary Table 4.

**Immunohistochemistry analyses**. Immunohistochemistry and immunofluorescence analyses were performed as previously described[66]. Briefly, dewaxed hydrated paraffin-embedded tissue sections were pre-incubated with 10% normal goat serum (1:10 dilution;#S-1000; Vector Laboratories for anti-MIG-6, Ki67, Cyclin D1, ErbB2, EGFR, pERK1/2, ERK1/2, MUC1, LTF, and KLF4 antibodies) or 10% normal rabbit serum (1:10 dilution; #S-5000; Vector Laboratories for anti-MCM2, MCM6, and KLF15 antibodies) serum in PBS and then incubated with anti-MIG-6 (1:200 dilution; Customized antibody by Dr. Jeong Lab), anti-Ki67 (1:1000 dilution; #ab15580; Abcam), anti-Cyclin D1 (1:1000 dilution; #eo-RB9041-p0; Thermo Fisher Scientific), anti-ErbB2 (1:200 dilution; #2165; Cell Signaling), anti-EGFR (1:200 dilution; #2646; Cell Signaling), anti-pERK1/2 (1:500 dilution; #4370; Cell Signaling), anti-ERK1/2 (1:1000 dilution; #4695; Cell Signaling), anti-MUC1 (1:1000 dilution; #ab15481, Abcam), anti-LTF (1:2000 dilution; #07-682, Millipore Corp.), anti-MCM2 (1:20000 dilution; #sc9839, Santa Cruz Biotechnology), anti-MCM6 (1:20000 dilution; #sc9843; Santa Cruz Biotechnology), anti-KLF4 (1:5000 dilution; #sc20691; Santa Cruz Biotechnology), and anti-KLF15 (1:5000 dilution; #ab2647; Abcam) antibodies in PBS supplemented with 10% normal serum overnight at 4 °C. For immunohistochemistry, the sections were incubated with secondary antibody conjugated to horseradish peroxidase (1:1000 dilution; #43-4324; Invitrogen) for one hour at room temperature. Immunoreactivity was detected using diaminobenzidine (DAB-Vector Laboratories) and analyzed using microscopy software from NIS Elements, Inc. (Nikon). Isotype control (anti-mouse (1:500 dilution; #BA-9200; Vector Laboratories), anti-rabbit (1:500 dilution; #BA-1000; Vector Laboratories), or anti-goat (1:2000 dilution; #BA-9500; Vector Laboratories) IgG) or no primary antibody control for IHC were used for negative controls (Supplementary Fig. 11). A semi-quantitative grading system (H-score) was calculated to compare the immunohistochemical staining intensities. The H-score was calculated using the following equation: H-score = $\sum$ Pi (i), where i=intensity of staining with a value of 1, 2, or 3 (weak,

moderate, or strong, respectively) and Pi is the percentage of stained cells for each intensity, varying from 0 to 100%. The overall score ranged from 0 to 300[67].

**Western blot analysis**. Western blot analyses were performed as described previously[68]. Proteins were extracted using lysis buffer (10 mM Tris-HCl (pH 7.4), 150 mM NaCl, 2.5 mM EDTA, and 0.125% Nonidet P-40 (vol/vol)) supplemented with both a protease inhibitor cocktail (Roche, Indianapolis, IN) and a phosphatase inhibitor cocktail (Sigma-Aldrich, St. Louis, MO). Protein lysates were electrophoresed via SDS-PAGE and transferred onto polyvinylidene difluoride membrane (Millipore Corp., Bedford, MA). Membrane was blocked with Casein (0.5% w/v) in PBS with 0.1% Tween 20 (v/v; Sigma-Aldrich) prior to exposure to anti-ErbB2 (1:1000 dilution; #2165; Cell Signaling, Danvers, MA), anti-EGFR (1:1000 dilution; #2646; Cell Signaling), anti-phospho-ERK1/2 (pERK1/2; 1:1000 dilution; #4370; Cell Signaling), anti-ERK1/2 (1:1000 dilution; #4695; Cell Signaling), anti-MIG-6 (1:1000 dilution; Customized antibody by Dr Jeong Lab) or anti-β-actin (1:1000 dilution; #sc1616; Santa Cruz Biotechnology) antibodies. Immunoreactivity was visualized by incubation with a horseradish peroxidase-linked secondary antibody (anti-mouse IgG (1:5000 dilution; Cat.# PI-2000; Vector Laboratories) or anti-rabbit IgG (1:5000 dilution; Cat.# PI-9500; Vector Laboratories) or anti-rabbit IgG (1:5000 dilution; Cat.#PI-1000; Vector Laboratories) followed by exposure to Electrochemiluminescence reagents (ECL) according to the manufacturer's instructions (GE Healthcare Biosciences).

**Statistical analysis**. No statistical method was used to predetermine sample size for in vivo studies. Based on prior experience, all experiments used five mice per group to achieve adequate statistical power. For all animal experiments, block randomization was used to ensure a balance in sample size across groups. The investigators were blinded during the evaluation of results variations in the group. For all animal experiments, over three biological replicates were analyzed for each condition, and results are presented as the mean ± SEM. For data with only two groups, two-tailed unpaired t test was used. For data containing more than two groups, an analysis of Ordinary one-way variance (ANOVA) test was used, followed by Tukey test for pairwise t test. $p < 0.05$ was considered statistically significant. All statistical analyses were performed using the Prism version 9.2.0 from GraphPad. Source data are provided in the Source Data file.

**Reporting summary**. Further information on research design is available in the Nature Research Reporting Summary linked to this article.

## Data availability

The microarray analysis data generated in this study have been deposited in the NCBI Gene Expression Omnibus database under accession code GSE138185.

All data generated in this study are provided in the Supplementary Information/Source Data file.

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

## Acknowledgements

This work was supported by the National Research Foundation of Korea (NRF) grant funded by the Korean Government (MIST) (No. 2018R1A5A2025079 and 2020R1A2C3003303 to H.-G.Y.), and SRI and Bayer Discovery/Innovation Grant (to T.H.K.), as well as by the Eunice Kennedy Shriver National Institute of Child Health & Human Development of the National Institutes of Health under Award Numbers R01HD084478, R01HD101243, and R01HD102170 (to J.W.J) and F31HD101207 and T32HD087166 (to R.M.M.). The content is solely the responsibility of the authors and does not necessarily represent the official views of the National Institutes of Health.

## Author contributions

H.-G.Y. and J.-W.J. were responsible for the concept of the study; A.T.F. collected baboon samples; S.L.Y. and B.A.L. collected human samples; J.-Y.Y. and T.H.K. carried out experiments; J.-Y.Y., T.H.K., and J.-H.S. analyzed data; U.M. provided transgenic mice; R.M.M. contributed to writing the manuscript. All authors contributed to the final manuscript version.

## Competing interests

The authors declare no competing interests.
