## [Peer Review File · Nature Communications]

REVIEWER COMMENTS

Reviewer #1 (Remarks to the Author):

This is a clearly written manuscript addressing the issue of endometrial progesterone resistance. The main objective is to provide data on role of MIG-6 in endometrial progesterone-resistance and thereby contribute to understanding further the molecular mechanisms responsible for endometrial progesterone resistance, notably in the context of endometriosis.

It is very interesting to see these data identifying the impact of disturbed progesterone signalling upon epithelial proliferation, rather than solely decidualization: the latter so often the case.

Data are presented from human endometrial biopsies and two animal models (non-human primate (NHP, baboon) and mouse) permitting interrogation of endometrial responses to modulation of a progesterone-induced gene, MIG-6. In endometrium from infertile women with endometriosis, NHP model of endometriosis, MIG-6, is downregulated. In contrast, in mice modeled with endometriosis and with Mig-6 deficient endometrium an increase of endometriosis development and implantation failure is described.

Observational data in Figure 1 present the case for the association of loss of MIG-6 and the presence of endometriosis in women and the NHP (baboon).

More incisive data are then shared from a mouse model: in uterine-specific Mig-6 knock-out mice data are presented to demonstrate that MIG-6 loss results in endometrial progesterone resistance via ERBB2. Specifically, Mig-6 loss exacerbates development of endometriosis and implantation failure in the mouse model and there is an associated increased expression in ERBB2. When there is additional ablation of ERBB2 the phenotype is rescued. The supplementary data are useful inclusions.

The microarray studies reveal genes observed to be differentially expressed in Mig-6d/d mice reverted to normal expression amounts with additional Erbb2 ablation in Mig-6d/dErbb2d/d mice.

The authors conclude that their data support MIG-6-induced ERBB2 overexpression contributing to to endometrial progesterone resistance and thus a role in endometriosis-related infertility. In their mouse model, targeting ERBB2 reverses the phenotype.

Significance to the field:

From a clinical perspective the focus is on endometriosis and mechanisms underpinning associated female subfertility where there are many lines of evidence suggesting progesterone resistance. Usually the impact of progesterone resistance is reported against a context of impact upon decidualization: herein there are very interesting data presented concerning the impact on epithelial proliferation, and consequently endometrial function of relevance to fertility/ subfertility.

Context with published literature:

The introduction is succinct presenting the case of endometrial progesterone-resistance as a phenomenon in women with endometriosis. This is supported with published data from animal models. The dearth of data concerning mechanisms involved is highlighted and thus where the present data make a highly important contribution. The data build further upon earlier work that indicate Mig-6 is a progesterone-regulated gene that mediates progesterone repression of estrogen action in the mouse uterus (Jeong JW, et al. *Endocrinology*, 2005; Jeong JW, et al. *Proc Natl Acad Sci*, 2009).

Methodology:

Research ethical approvals are in place for human and animal studies.

The "phenotyping" of participants is a crucial component of this study pertaining to progesterone-resistance in women with endometriosis.

The dating of the endometrium from women with and without endometriosis is based upon histology. If available, further support for important menstrual cycle staging would be provided by

serum sex hormone (estradiol, progesterone) at the time of endometrial biopsy.

Please clarify if any other commonly present uterine myometrial confounders, also reported to be associated with potential progesterone-resistance, i.e. small uterine leiomyoma and adenomyosis were present in any of the participants from whom endometrium was sampled.

RT-q-PCR and immunohistochemistry (IHC) are standard.

For the IHC please clarify negative controls employed.

For the microarray studies: how was quality of RNA established?

Results/ figures:

Circulating serum progesterone concentrations are highest in the early and mid-secretory (ES, MS) phases with a physiological decline in notably progesterone in the late-secretory (LS) phase. In Figure 1A it is noted MIG-6 expression is highest in the ES: please comment further here as if a progesterone- induced gene - would not levels also be expected to be higher in MS stage?

In Figure 1A even though in the figure legend it is stated $n > 3$ per stage please give $n =$ per stage.

In Figure 1B please clarify in legend the stage of the menstrual cycle H-scores were measured.

Please enhance legends of figures 2B; 2C; 4E to explain the fluorescence photomicrographs.

Figure 3: provides very clear support in the mouse for the impact upon epithelial proliferation and rescue of implantation.

Figure 6: depicts the proposed molecular mechanisms of MIG-6 function in the uterus. This summarises the important contributions in the manuscript and notably the impact upon epithelial proliferation. The further pathways/ mechanisms precisely responsible for subfertility in women with endometriosis still require to be established: hence this should be made clear.

Supplementary data illustrate an impressive reversal of endometrial epithelial hyperplasia.

Reviewer #2 (Remarks to the Author):

In endometriosis patients, overall P4-effects on the endometrium are diminished (P4 resistance). MIG-6 is one of the P4-induced genes in the endometrium, mediating anti-proliferation effects of P4 on the epithelial cells. In this study, the authors investigate the functions of MIG-6 in the pathogenesis of endometriosis and endometriosis-associated fertility problems. The authors discovered that MIG-6 expression was reduced in the endometrium of infertile women diagnosed with endometriosis. In the induced-endometriosis models of non-human primates and mice, MIG-6 expression in the eutopic endometrium progressively decreased as endometriosis lesions grew. In the mouse endometrium, the deletion of Mig-6 by Pgr-Cre blunted the epithelial response to P4, but the co-deletion of Erbb2 corrected the epithelial defects. Therefore, the authors concluded that MIG-6 downregulation and subsequent ERBB2 upregulation contribute to the P4 resistance of the endometriotic endometrium.

The authors have already shown that MIG-6 is a crucial mediator of P4 action in the endometrium by utilizing the same mouse model. Thus, this study's primary finding is that P4 represses endometrial epithelial proliferation in part by repressing ERBB2 expression via upregulation of MIG-6.

Given P4 resistance is one of the hallmarks, all P4-regulated genes, including MIG-6, are expected to be downregulated in the eutopic and ectopic endometria of endometriotic women. The current study confirms this idea. MIG-6 is a negative regulator of EGF pathways known to reduce the expression levels EGFRs, including ERBB2, by promoting internalization and degradation via binding to the intracellular domain. Thus, MIG-6 downregulation and subsequent ERBB2 overexpression confirm the well-accepted molecular functions of MIG-6.

Overall, this study supports the well-accepted concept that the loss of P4 pathway activity causes endometriosis. The impact would have been higher if this study provided insights into the mechanism of MIG-6 downregulation in the endometrium by endometriosis progression. The study's impact would also increase by testing if ERBB2 null mutation can rescue PGR-null mice's endometrial phenotypes.

The fundamental role of P4 in the endometrium is to counteract against E2 actions. The authors should have analyzed the effects of E2 alone in the endometrium of normal and mutant mice. Without this baseline, the P4 effect cannot be accurately evaluated. The expression patterns of PGR and ESR1 in the uterus under different hormonal conditions should also be compared among mice in different genotypes.

The inclusion of *ErbB2* d/d mice would help understand the role of MIG-6 and ERBB2 in normal uterine functions.

Since *Pgr-Cre* is expressed in the ovary, the reproductive phenotypes of *Mig-6* d/d and *Mig6* d/d *ErbB2* d/d mice cannot be attributed solely to the uterine defects. The ovarian phenotypes of knockout mice should be included in the study.

The recombination efficiency of *Pgr-Cre* is not 100%. Incomplete deletion would increase when 4 floxed alleles are simultaneously targeted. To establish the dispensability of MIG6 and ERBB2 in implantation, the loss of ERBB2 and MIG-6 in the implantation sites of double knockout mice should be tested by IHC.

Figure S2 indicates mice who received the endometriosis-inducing surgery showed two distinct responses; the decidual response was retained and lost. The authors should analyze the correlations between the number of implantation sites and the number/size of endometriosis lesions.

Figure 2B. As assessed by the brightness in the areas with no tissues (e.g., epithelial lumen and large vessels), the background signal levels are significantly different among pictures. How did the authors adjust the background levels in different samples? The methods do not describe the adjustment of the background.

Fig. S3. The finding that EGFR is not overexpressed in the uterus of MIG-6 d/d mice is intriguing. The difference between EGFR and ERBB2 likely reflects the difference in the tissues expressing these receptors. The authors may consider including EGFR IHC.

Figure 3 analyzes the expression levels of epithelial genes. As Figure 3B shows, the epithelium amount is significantly higher in *Mig-6* d/d mice than the other two genotypes. The analyses would be more accurate if the transcript levels are normalized against epithelial-genes, such as E-cadherin and Keratin-18.

The rationale of the experiment presented in Figure 4 D and E is unclear. P4 resistance, including the downregulation of MIG-6, is a consequence of endometriosis development, as shown in Figure 1 D and E. Do the authors propose an alternative model in which P4 resistance causes endometriosis?

Reviewer #3 (Remarks to the Author):

This study implicates MIG-6 as a critical mediator of endometriosis that functions through ERBB2. The study describes and refines several novel animal models and represents a major new contribution to the field of endometriosis research. The rescue of MIG-6 phenotypes by simultaneous ERBB2 inactivation is a stunning and elegant genetic result that strongly substantiates the scientific claims made in this study. The authors deserve credit for this result and also for the inclusion of data from human specimens and also from a non-human primate model of

endometriosis that together make for a very strong study that clearly merits publication in Nature Communications. Also, these results are a significant advance over prior work on MIG-6 that has largely been limited to mouse genetic models. The manuscript is well-written and the results clearly presented, and the conclusions well-substantiated by the data. The supplementary data is of very high quality and adds to the study. However, there are some minor technical and scientific issues that merit some attention prior to publication:

Introduction-clear and succinct, but the authors should present more background information on MIG-6 for a general readership; e.g. type of protein, regulation, biological context, etc.

Fig. 1. Does the gradual loss of MIG-6 in the non-human primate model correlate with important biological parameters (proliferation rate by Ki67, secretory changes, etc.)?

Fig. 2. Since panel A represents quantitation of the images in panel b, it might make more sense to switch panel order.

Fig. 4 and elsewhere. It would be worthwhile for the authors to present data on ER and PR expression in their various models as this could be related to the underlying biology and observations.

Finally, is ERBB2 expression aberrant in the human and primate models?

We sincerely thank the editor and the reviewers for carefully reviewing this manuscript and for the insightful comments, which have helped improve the quality of the manuscript. We are pleased with the positive comments of the reviewers.

The Editorial Requests

POLICIES AND FORMS REQUIRED FOR RESUBMISSION

We have uploaded “Editorial policy checklist” and “Reporting summary”.

DATA AND CODE AVAILABILITY

Our microarray datasets have been deposited to Gene Expression Omnibus (accession code: GSE138185). It is described in “Data Availability” section.

To maximize the reproducibility of research data, we have provided an excel file containing the raw data.

We also provided uncropped versions of blots for Supplementary Figures 4 and 5 in Supplementary Figures 13.

We have replaced our bar graphs with plots that feature information about the distribution of the underlying data.

ORCID

We provided ORCID for all authors.

Reviewer #1

Overall Comment

This is a clearly written manuscript addressing the issue of endometrial progesterone resistance. The main objective is to provide data on role of MIG-6 in endometrial progesterone-resistance and thereby contribute to understanding further the molecular mechanisms responsible for endometrial progesterone resistance, notably in the context of endometriosis.

It is very interesting to see these data identifying the impact of disturbed progesterone signalling upon epithelial proliferation, rather than solely decidualization: the latter so often the case.

Data are presented from human endometrial biopsies and two animal models (non-human primate (NHP, baboon) and mouse) permitting interrogation of endometrial responses to modulation of a progesterone-induced gene, MIG-6. In endometrium from infertile women with endometriosis, NHP model of endometriosis, MIG-6, is downregulated. In contrast, in mice modeled with endometriosis and with Mig-6 deficient endometrium an increase of endometriosis development and implantation failure is described.

Observational data in Figure 1 present the case for the association of loss of MIG-6 and the presence of endometriosis in women and the NHP (baboon).

More incisive data are then shared from a mouse model: in uterine-specific Mig-6 knock-out mice data are presented to demonstrate that MIG-6 loss results in endometrial progesterone resistance via ERBB2. Specifically, Mig-6 loss exacerbates development of endometriosis and implantation failure in the mouse model and there is an associated increased expression in ERBB2. When there is additional ablation of ERBB2 the phenotype is rescued. The supplementary data are useful inclusions.

The microarray studies reveal genes observed to be differentially expressed in Mig-6d/d mice reverted to normal expression amounts with additional Erbb2 ablation in Mig-6d/dErbb2d/d mice.

The authors conclude that their data support MIG-6-induced ERBB2 overexpression contributing to endometrial progesterone resistance and thus a role in endometriosis-related infertility. In their mouse model, targeting ERBB2 reverses the phenotype.

Significance to the field:

From a clinical perspective the focus is on endometriosis and mechanisms underpinning associated female subfertility where there are many lines of evidence suggesting progesterone resistance. Usually the impact of progesterone resistance is reported against a context of impact upon decidualization: herein there are very interesting data presented concerning the impact on epithelial proliferation, and consequently endometrial function of relevance to fertility/ subfertility.

Context with published literature:

The introduction is succinct presenting the case of endometrial progesterone-resistance as a phenomenon in women with endometriosis. This is supported with published data from animal models. The dearth of data concerning mechanisms involved is highlighted and thus where the present data make a highly important contribution. The data build further upon

earlier work that indicate Mig-6 is a progesterone-regulated gene that mediates progesterone repression of estrogen action in the mouse uterus (Jeong JW, et al. Endocrinology, 2005; Jeong JW, et al. Proc Natl Acad Sci ,2009).

Response: We are pleased with the positive comment.

Methodology:

C1. Research ethical approvals are in place for human and animal studies.
The “phenotyping” of participants is a crucial component of this study pertaining to progesterone-resistance in women with endometriosis.

R1. Thank you

C2. The dating of the endometrium from women with and without endometriosis is based upon histology. If available, further support for important menstrual cycle staging would be provided by serum sex hormone (estradiol, progesterone) at the time of endometrial biopsy.

R2. Unfortunately, we do not have the levels of serum estradiol and progesterone at the time of endometrial biopsy. However, histologic dating of endometrial samples was done on the basis of the criteria of Noyes et al. and confirmed by subsequent histopathological examination by an experienced fertility specialist (B.A.L.).

C3. Please clarify if any other commonly present uterine myometrial confounders, also reported to be associated with potential progesterone-resistance, i.e. small uterine leiomyoma and adenomyosis were present in any of the participants from whom endometrium was sampled.

R3. None of the participating patients had known uterine leiomyoma or adenomyosis.

C4. RT-q-PCR and immunohistochemistry (IHC) are standard.
For the IHC please clarify negative controls employed.
For the microarray studies: how was quality of RNA established?

R4. We employed negative controls including isotype control (IgG) or no primary antibody control for IHC (see Suppl Fig. 01). NanoDrop was used to determine RNA purity and for an initial estimate of RNA concentration. All RNA samples were analyzed with a Bioanalyzer 2100 (Agilent Technologies, Wilmington, DE) to confirm sample concentration and purity (RIN > 8.0; concentration 100-200ng/ul) before microarray hybridization (See Line 450 - 452).

Results/ figures:

C5. Circulating serum progesterone concentrations are highest in the early and mid-secretory (ES, MS) phases with a physiological decline in notably progesterone in the late-secretory (LS) phase. In Figure 1A it is noted MIG-6 expression is highest in the ES: please comment further here as if a progesterone- induced gene - would not levels also be expected to be higher in MS stage?

R5. This is a very important comment. We used transcriptomic analysis to identify alterations in gene expression after acute and chronic P4 treatment in the mouse uterus (PMID: 15845616). *Mig-6* was identified as a rapidly induced gene by acute P4 treatment (4 hours), but its induction was not continued with chronic P4 treatment (40 hours). As we observed in the mouse uterus, the expression of MIG-6 was significantly increased in human endometrium at early secretory phase compared to proliferative phase, but this induction was not observed at Mid-secretory phase. These results suggest that MIG-6 is an acute P4 response gene in both the human and mouse endometrium.

C6. In Figure 1A even though in the figure legend it is stated $n > 3$ per stage please give $n =$ per stage.

R6. We have provided exact sample number for Fig. 1A. In addition, the bar graph has been replaced with dot plots to clearly show the sample number as well as the data distribution.

C7. In Figure 1B please clarify in legend the stage of the menstrual cycle H-scores were measured.

R7. The samples were from early secretory phase women.

C8. Please enhance legends of figures 2B; 2C; 4E to explain the fluorescence photomicrographs.

R8. We have provided more detailed information in Figure Legends 3B, 3C, and 6E Figure Legends (Former 2B, 2C, and 4E).

C9. Figure 3: provides very clear support in the mouse for the impact upon epithelial proliferation and rescue of implantation.

R9. Thank you for this encouraging comment.

C10. Figure 6: depicts the proposed molecular mechanisms of MIG-6 function in the uterus. This summarises the important contributions in the manuscript and notably the impact upon epithelial proliferation. The further pathways/ mechanisms precisely responsible for subfertility in women with endometriosis still require to be established: hence this should be made clear.

R10. We agree with the reviewer. We have added this comment for the further study in the Discussion.

“Figure 8 depicts our proposed molecular mechanisms of MIG-6 function in endometrial P4 resistance and associated infertility and how P4 responsiveness and fertility can be restored. However, the further pathways and mechanisms precisely responsible for subfertility in women with endometriosis still require to be established.” (See Line 336 - 339).

Reviewer #2

Overall Comment

In endometriosis patients, overall P4-effects on the endometrium are diminished (P4 resistance). MIG-6 is one of the P4-induced genes in the endometrium, mediating anti-proliferation effects of P4 on the epithelial cells. In this study, the authors investigate the functions of MIG-6 in the pathogenesis of endometriosis and endometriosis-associated fertility problems. The authors discovered that MIG-6 expression was reduced in the endometrium of infertile women diagnosed with endometriosis. In the induced-endometriosis models of non-human primates and mice, MIG-6 expression in the eutopic endometrium progressively decreased as endometriosis lesions grew. In the mouse endometrium, the deletion of Mig-6 by Pgr-Cre blunted the epithelial response to P4, but the co-deletion of Erbb2 corrected the epithelial defects. Therefore, the authors concluded that MIG-6 downregulation and subsequent ERBB2 upregulation contribute to the P4 resistance of the endometriotic endometrium.

The authors have already shown that MIG-6 is a crucial mediator of P4 action in the endometrium by utilizing the same mouse model. Thus, this study's primary finding is that P4 represses endometrial epithelial proliferation in part by repressing ERBB2 expression via upregulation of MIG-6.

Given P4 resistance is one of the hallmarks, all P4-regulated genes, including MIG-6, are expected to be downregulated in the eutopic and ectopic endometria of endometriotic women. The current study confirms this idea. MIG-6 is a negative regulator of EGF pathways known to reduce the expression levels EGFRs, including ERBB2, by promoting internalization and degradation via binding to the intracellular domain. Thus, MIG-6 downregulation and subsequent ERBB2 overexpression confirm the well-accepted molecular functions of MIG-6.

Overall, this study supports the well-accepted concept that the loss of P4 pathway activity causes endometriosis.

Response: We are pleased with the positive comment.

C1. The impact would have been higher if this study provided insights into the mechanism of MIG-6 downregulation in the endometrium by endometriosis progression.

R1. We found that the attenuation of MIG-6 in endometriosis and MIG-6 loss causes infertility due to an implantation failure in the mouse uterus. Our animal models suggest that endometriosis results in MIG-6 loss in the eutopic endometrium of women with endometriosis. Endometriosis changes immune function, cytokines and inflammatory signals that may lead to progesterone resistant conditions including MIG-6 loss in the eutopic endometrium. However, we could not demonstrate the mechanism of MIG-6 loss in the endometrium over endometriosis progression. Identification of the mechanism how endometriosis leads to MIG-6 downregulation in the endometrium is important to understand the pathophysiology of progesterone resistance in endometriosis-related infertility.

C2. The study's impact would also increase by testing If ERBB2 null mutation can rescue PGR-null mice's endometrial phenotypes.

R2. Taking the Reviewers' advice, we have generated *Pgr^{cre/cre}Erb2^{ff}* and *Pgr^{cre/f}Erb2^{ff}* mice to determine the effect of ERBB2 ablation in PRKO mice. However, ERBB2 null mutation did not rescue the phenotype of implantation failure in PRKO mice (see Suppl. Fig. 08).

C3. The fundamental role of P4 in the endometrium is to counteract against E2 actions. The authors should have analyzed the effects of E2 alone in the endometrium of normal and mutant mice. –Without this baseline, the P4 effect cannot be accurately evaluated. Fig. 1E; Fig. 3C

R3. Taking Reviewer's suggestion, ovariectomized control, *Mig-6^{d/d}*, *Erb2^{d/d}*, and *Mig-6^{d/d}Erb2^{d/d}* mice were treated with E2 alone for 3 days. However, the uterus/body weight measurements were not different between control, *Mig-6^{d/d}*, *Erb2^{d/d}*, and *Mig-6^{d/d}Erb2^{d/d}* mice after E2 treatment (Suppl. Fig. 09). Therefore, we have not included the E2 alone condition in molecular analysis.

C4. The expression patterns of PGR and ESR1 in the uterus under different hormonal conditions should also be compared among mice in different genotypes.

R4. In previous study, we showed that the expressions of PGR and ESR1 were not changed in the uteri of *Mig-6^{d/d}* mice compared to control mice at GD 3.5. As we expected, the expression levels of PGR and ESR1 were not changed in the uteri of *Erb2^{d/d}* and *Mig-6^{d/d}Erb2^{d/d}* mice compared to control mice at GD 3.5 (Suppl. Fig. 06).

C5. The inclusion of *Erb2 d/d* mice would help understand the role of MIG-6 and ERBB2 in normal uterine functions.

R5. Taking the Reviewer's advice, we have included *Erb2^{d/d}* mice in Figure 4, 5, 6, S6, S7, S9 and S11. *Erb2^{d/d}* mice show normal uterine phenotypes like control mice.

C6. Since *Pgr-Cre* is expressed in the ovary, the reproductive phenotypes of *Mig-6 d/d* and *Mig6 d/d Erb2 d/d* mice cannot be attributed solely to the uterine defects. The ovarian phenotypes of knockout mice should be included in the study.

R6. We have reported that *Mig-6^{d/d}* mice did not show any alteration of ovarian function (PMID: 20018910). In addition, the number of implantation sites at GD5.5 of *Mig-6^{d/d}Erb2^{d/d}* and *Erb2^{d/d}* mice were same to control (Figure 4A). These data indicate that *Mig-6^{d/d}Erb2^{d/d}* and *Erb2^{d/d}* mice have a normal ovarian function.

C7. The recombination efficiency of *Pgr-Cre* is not 100%. Incomplete deletion would increase when 4 floxed alleles are simultaneously targeted. To establish the dispensability of MIG6

and ERBB2 in implantation, the loss of ERBB2 and MIG-6 in the implantation sites of double knockout mice should be tested by IHC.

R7. We have confirmed the loss of MIG-6 and ERBB2 expression in the uteri of *Mig-6^{d/d}* and *Mig-6^{d/d} Erbb2^{d/d}* mice at GD3.5 by IHC, RT-qPCR and Western blot analyses as shown in Figure 4, S4, and S5. These data support appropriate recombination efficiency of Pgr-Cre.

C8. Figure S2 indicates mice who received the endometriosis-inducing surgery showed two distinct responses; the decidual response was retained and lost. The authors should analyze the correlations between the number of implantation sites and the number/size of endometriosis lesions.

R8. This is a very insightful comment. We have examined the possible correlation between the number of implantation sites and the number of endometriosis lesions. However, no significant correlation exists between the number of implantation sites and the number of endometriosis lesions (Suppl. Fig. 03).

C9. Figure 2B. As assessed by the brightness in the areas with no tissues (e.g., epithelial lumen and large vessels), the background signal levels are significantly different among pictures. How did the authors adjust the background levels in different samples? The methods do not describe the adjustment of the background.

R9. The images were taken with exactly the same parameters, including exposure time, and the background of images was not adjusted. We think the background is darker in some images because MIG-6 proteins are expressed in cytoplasm.

C10. Fig. S3. The finding that EGFR is not overexpressed in the uterus of MIG-6 d/d mice is intriguing. The difference between EGFR and ERBB2 likely reflects the difference in the tissues expressing these receptors. The authors may consider including EGFR IHC.

R10. Taking the Reviewer's advice, we have included EGFR IHC in Fig. S4. As in the Western blot result, the results of EGFR IHC show that level of EGFR was not changed in in the uteri of *Mig-6^{d/d}* mice.

C11. Figure 3 analyzes the expression levels of epithelial genes. As Figure 3B shows, the epithelium amount is significantly higher in Mig-6 d/d mice than the other two genotypes. The analyses would be more accurate if the transcript levels are normalized against epithelial-genes, such as E-cadherin and Keratin-18.

R11. Taking the Reviewer's advice, we have confirmed our RT-qPCR results by normalizing against E-cadherin gene expression (see below results). In addition, there should not be many epithelial cells in normal post-implantation stage (gestation day 5.5) control mice (Fig. 4B, a and b) and *Mig-6^{d/d}Erbb2^{d/d}* mice (Fig. 4B, e and f). However, the phenotype of implantation failure in *Mig-6^{d/d}* mice revealed more epithelial cells due to no luminal closure at GD 5.5.

18S

E-Cad

C12. The rationale of the experiment presented in Figure 4 D and E is unclear. P4 resistance, including the downregulation of MIG-6, is a consequence of endometriosis development, as shown in Figure 1 D and E. Do the authors propose an alternative model in which P4 resistance causes endometriosis?

R12. Yes, we determined the effect of MIG-6 loss on endometriosis development. According to Sampson's theory of retrograde menstruation, menstrual blood containing endometrial cells flows backward through the fallopian tubes and into the pelvic cavity rather than out of the body. These endometrial cells that should have been shed during menstruation can then lead to implantation and further spreading of endometriosis lesions. We found that *Mig-6^{d/d}Rosa26^{mTmG/+}* mice had a significantly increased incidence of endometriotic lesions compared to control mice. Therefore, MIG-6 loss in eutopic endometrium by endometriosis accelerates progress of this disease.

Reviewer #3

Overall Comment

This study implicates MIG-6 as a critical mediator of endometriosis that functions through ERBB2. The study describes and refines several novel animal models and represents a major new contribution to the field of endometriosis research. The rescue of MIG-6 phenotypes by simultaneous ERBB2 inactivation is a stunning and elegant genetic result that strongly substantiates the scientific claims made in this study. The authors deserve credit for this result and also for the inclusion of data from human specimens and also from a non-human primate model of endometriosis that together make for a very strong study that clearly merits publication in Nature Communications. Also, these results are a significant advance over prior work on MIG-6 that has largely been limited to mouse genetic models. The manuscript is well-written and the results clearly presented, and the conclusions well-substantiated by the data. The supplementary data is of very high quality and adds to the study. However, there are some minor technical and scientific issues that merit some attention prior to publication:

Response: We are pleased with the positive comment. We have tried our best to address minor technical and scientific issues in the manuscript.

C1. Introduction-clear and succinct, but the authors should present more background information on MIG-6 for a general readership; e.g. type of protein, regulation, biological context, etc.

R1. We have added more detailed information on MIG-6 to the Introduction.

C2. Fig. 1. Does the gradual loss of MIG-6 in the non-human primate model correlate with important biological parameters (proliferation rate by Ki67, secretory changes, etc.)?

R2. We observed increased ERBB2 in the eutopic endometrium from the same baboons over time and with endometriosis progression. There are reverse correlations between MIG-6 and ERBB2 proteins in Figure 6C.

C3. Fig. 2. Since panel A represents quantitation of the images in panel b, it might make more sense to switch panel order.

R3. Taking the Reviewer's advice, we have switched the panel order in Fig. 3.

C4. Fig. 4 and elsewhere. It would be worthwhile for the authors to present data on ER and PR expression in their various models as this could be related to the underlying biology and observations.

R4. Taking the Reviewer's advice, we have examined the levels of PGR and ESR1 in the uteri from our mutant mice at GD 3.5 (Suppl. Fig. 6) as well as ovariectomized mice treated with E2 plus P4 for 3 days (Suppl. Fig. 7). The results have been added to the manuscript.

C5. Finally, is ERBB2 expression aberrant in the human and primate models?

R5. This is a very important question. We have examined the expression of ERBB2 in the induced endometriosis of non-human primates. We observed increased ERBB2 in the eutopic endometrium from the same baboons over time and with endometriosis progression. There are reverse correlations between MIG-6 and ERBB2 proteins in Figure 6.

REVIEWERS' COMMENTS

Reviewer #1 (Remarks to the Author):

As a previous reviewer I consider the points I raised in review have been addressed.

I have a further minor comment: Concerning: Page 30 from 42 (in combined pdf) lines 731-4 Figure 1. MIG-6 expression in the endometrium of women with endometriosis and nonhuman primate, baboon model. (A), RT-qPCR analysis of MIG-6 gene expression in endometrium from women with and without endometriosis during the menstrual cycle ($n \geq 3$ for each group). Further clarity in legend is required as two sets of RT-qPCR analysis of MIG-6 gene expression are presented.

Overall: the major finding in current manuscript is an important addition to the field, i.e. that progesterone represses endometrial epithelial proliferation, involving ERBB2 expression, via upregulation of MIG-6. These current data presented build further upon earlier work that indicate Mig-6 is a progesterone-regulated gene that mediates progesterone repression of estrogen action in the mouse uterus.

As noted in my previous review, usually the impact of progesterone resistance is reported in the context of impact upon decidualization. These are very interesting data concerning the impact on epithelial proliferation, and consequently endometrial function of relevance to fertility/ subfertility.

Reviewer #2 (Remarks to the Author):

The authors satisfactorily addressed the critiques of the reviewers. Nevertheless, this reviewer recommends a minor revision as new data in the revised manuscript suggest a novel mechanism that has not been discussed before; the PGR-MIG6-ERBB2 signaling pathway makes a positive feedback loop. In previous studies, the authors showed P4 signal transduction from PGR to ERBB2 via MIG6. Surprisingly, the loss of MIG6 repressed the expression of PGR in both uterine epithelial and stromal cells when mice were treated with E2+P4 for 3 days (Figure S7). This observation indicates that the repression of ERBB2 by MIG6 potentiates P4 action on the uterine cells by maintaining the PGR expression. Therefore, the authors should discuss the potential significance of the feedback loop in pregnancy and endometriosis. In addition, the loss of ERBB2 restored PGR expression in both epithelial and stromal cells of the MIG6 null uterus, although ERBB2 expression is restricted to the epithelium. This observation confirms the notion that epithelial-stromal tissue communications regulate PGR in the mouse uterus (PMID: 10727249, PMID: 26858409). Thus, the discussion will be deepened if the authors address the role of cross-talk between epithelial and stromal cells in the PGR-MIG6-ERBB2 signaling pathway. This reviewer also recommends the authors discuss if the PGR-MIG6-ERBB2 signaling loop is conserved in the primate endometrium because PGR regulation in the primate endometrium (rhesus macaque [PMID: 15781981] and human PDX [PMID: 16138832]) is distinct from mice.

Reviewer #3 (Remarks to the Author):

The authors should be commended for their thorough responses to my criticisms and those of the other reviewers. Substantial changes were made to the manuscript, with new data and improvements in the presentation where appropriate. In my opinion, the revised manuscript is now suitable for publication in Nature Communications.

We sincerely thank the editor and the reviewers for carefully reviewing this manuscript. The insightful comments have helped improve the quality of the manuscript.

The Editorial Requests

We therefore invite you to revise your paper one last time to address the remaining concerns of our reviewers and our editorial requests in the attached documents. At the same time we ask that you edit your manuscript to comply with our policies and formatting requirements and to maximise the accessibility and therefore the impact of your work.

Please see the attached documents, listing a number of points that must be addressed. Failure to comply with our editorial requests will cause delays in accepting your manuscript. Please also see the Nature Communications formatting instructions for further information.

We have addressed all comments of “Author checklist” and “Reporting summary” and uploaded to the submission system.

Reviewer #1

As a previous reviewer I consider the points I raised in review have been addressed. I have a further minor comment:

C1. Concerning: Page 30 from 42 (in combined pdf) lines 731-4

Figure 1. MIG-6 expression in the endometrium of women with endometriosis and nonhuman primate, baboon model. (A), RT-qPCR analysis of MIG-6 gene expression in endometrium from women with and without endometriosis during the menstrual cycle ($n \geq 3$ for each group). Further clarity in legend is required as two sets of RT-qPCR analysis of MIG-6 gene expression are presented.

R1. Taking the Reviewers' advice, we have revised the figure legend.

“Figure 1. MIG-6 expression in the endometrium of women with endometriosis and nonhuman primate, baboon model. (a), RT-qPCR analysis of *MIG-6* gene expression in endometrium from women with and without endometriosis during the menstrual cycle ($n = 6$ for proliferative, $n = 7$ for early secretory, $n = 3$ for mid secretory, and $n = 6$ for late secretory without endometriosis and $n = 6$ for early secretory, $n = 9$ for mid secretory, and $n = 3$ for late secretory with endometriosis). Filled square boxes represent endometrial samples from women without endometriosis, and empty square boxes represent endometrial samples from women with endometriosis. Data are represented as mean \pm SEM, * $p=0.0106$, ** $p=0.0029$, *** $p<0.0001$, and ** $p=0.0079$ by Ordinary one-way ANOVA test. (b), (c), Immunohistochemical H-score (c) and representative photomicrographs (c) of MIG-6 in the endometrium from women with endometriosis as compared to controls ($n = 10$ for each group) at early secretory phase. Data are represented as mean \pm SEM, *** $p<0.0001$ by two-tailed unpaired t-test. (d), Immunohistochemical H-score and representative photomicrographs of MIG-6 in the endometriosis baboon model induced by intraperitoneal inoculation of menstrual endometrium during progression of endometriosis in pre-inoculation, 3, 6, and 9 months ($n = 4$ per period). Data are represented as mean \pm SEM, *** $p<0.0001$ by Ordinary one-way ANOVA test. (e), Immunohistochemical H-score and representative photomicrographs of MIG-6 in the endometriosis mouse model ($n = 5$ for group). Data are represented as mean \pm SEM, *** $p<0.0001$ by two-tailed unpaired t-test. Source data are provided in the Source Data file.”

C2. Overall: the major finding in current manuscript is an important addition to the field, i.e. that progesterone represses endometrial epithelial proliferation, involving ERBB2 expression, via upregulation of MIG-6. These current data presented build further upon earlier work that indicate Mig-6 is a progesterone-regulated gene that mediates progesterone repression of estrogen action in the mouse uterus.

As noted in my previous review, usually the impact of progesterone resistance is reported in the context of impact upon decidualization. These are very interesting data concerning the impact on epithelial proliferation, and consequently endometrial function of relevance to fertility/subfertility.

R2. Thank you very much for your positive view of our study.

Reviewer #2:

C1. The authors satisfactorily addressed the critiques of the reviewers. Nevertheless, this reviewer recommends a minor revision as new data in the revised manuscript suggest a novel mechanism that has not been discussed before; the PGR-MIG6-ERBB2 signaling pathway makes a positive feedback loop. In previous studies, the authors showed P4 signal transduction from PGR to ERBB2 via MIG6. Surprisingly, the loss of MIG6 repressed the expression of PGR in both uterine epithelial and stromal cells when mice were treated with E2+P4 for 3 days (Figure S7). This observation indicates that the repression of ERBB2 by MIG6 potentiates P4 action on the uterine cells by maintaining the PGR expression. Therefore, the authors should discuss the potential significance of the feedback loop in pregnancy and endometriosis.

R1. We appreciate this valuable comment. We have added the following content to the Discussion on lines 310- 322:

“Previously we identified MIG-6 as a target of PGR in the uterus necessary to retain P4-responsiveness for endometrial homeostasis and successful embryo implantation [PMID: 19439667, PMID: 20018910]. Past studies also showed that MIG-6 binds to ERBB2 and inhibits its signaling in vitro [PMID: 11003669, PMID: 12833145, PMID: 18046415]. In the present study, we delineate the action of PGR through MIG-6 to suppress ERBB2 expression in vivo in the uterus. Interestingly, we observed a reduction of PGR expression in *Mig-6^{d/d}* endometrial epithelium and stroma compared to controls after E2+P4 treatment for 3 days, and this change was rescued by the additional ablation of *ErbB2*. This finding raises the possibility of a positive feedback loop where repression of *ErbB2* by PGR-induced MIG-6 expression is necessary to maintain PGR expression in the presence of combined E2+P4 treatment. Therefore, the negative consequences of MIG-6 loss in the endometrium of infertile women with endometriosis may include further loss of PGR and P4-responsiveness. However, we did not observe the loss of PGR in *Mig-6^{d/d}* mice at GD3.5, implying that this feedback mechanism may not be active during natural pregnancy conditions.”

C2. In addition, the loss of ERBB2 restored PGR expression in both epithelial and stromal cells of the MIG6 null uterus, although ERBB2 expression is restricted to the epithelium. This observation confirms the notion that epithelial-stromal tissue communications regulate PGR in the mouse uterus (PMID: 10727249, PMID: 26858409). Thus, the discussion will be deepened if the authors address the role of cross-talk between epithelial and stromal cells in the PGR-MIG6-ERBB2 signaling pathway.

R2. Although expression of ERBB2 is not limited to epithelium, the mechanism of action of ERBB2 is thought to occur mainly in epithelium. In the reviewer's opinion, regulation of crosstalk between epithelial and stromal cells is considered a major role for the PGR-MIG6-ERBB2 signaling pathway. Functional regulation of ERBB2 by MIG-6 in epithelial cells is thought to be a key mechanism regulating P4-PGR signaling in stromal cells to carry out the normal function of the uterus.

We have added the following to the discussion on lines 322 - 327:

“In addition, it is interesting in light of known epithelial-stromal crosstalk mechanisms regulating endometrial PGR [PMID: 10727249] that ERBB2 overexpression in *Mig-6^{d/d}* uteri occurs primarily, though not only, in the

epithelium. However, PGR reduction after E2+P4 treatment occurs in both the epithelial and stromal compartments of *Mig-6^{d/d}* uteri, and it is rescued in both compartments by *ErbB2* deletion. These data imply the presence of epithelial to stromal communication in the feedback from ERBB2 overexpression to PGR abrogation.”

C3. This reviewer also recommends the authors discuss if the PGR-MIG6-ERBB2 signaling loop is conserved in the primate endometrium because PGR regulation in the primate endometrium (rhesus macaque [PMID: 15781981] and human PDX [PMID: 16138832]) is distinct from mice.

R3. We observed an inverse correlation between MIG-6 and ERBB2 proteins in eutopic endometrium from the same baboons over time with endometriosis progression. Of course, further experimentation is needed, but we think that the PGR-MIG6-ERBB2 signaling pathway is an important regulatory mechanism of uterine function in primates and humans.

We have added the following to the discussion on line 328 - 333:

“Since PGR regulation in human and non-human primate endometrial tissue is not identical to mouse tissue [PMID: 16138832] [PMID: 15781981], we cannot be certain whether this positive feedback mechanism is conserved between species. However, our data from a baboon endometriosis model shows a significant inverse correlation between MIG-6 and ERBB2 in the eutopic endometrium. As MIG-6 decreases with endometriosis progression, ERBB2 increases, which supports the relevance of this pathway to uterine dysfunction in primates.”

Reviewer #3:

C1. The authors should be commended for their thorough responses to my criticisms and those of the other reviewers. Substantial changes were made to the manuscript, with new data and improvements in the presentation where appropriate. In my opinion, the revised manuscript is now suitable for publication in Nature Communications.

R1. Thank you very much for the invaluable comments.